# Fibrogranular materials function as organizers to ensure the fidelity of multiciliary assembly

Huijie Zhao [1,2,7], Qingxia Chen[1,3,7], Fan Li[1,2], Lihong Cui[4], Lele Xie[1], Qiongping Huang[1], Xin Liang [4], Jun Zhou [5], Xiumin Yan[1✉] & Xueliang Zhu [1,2,3,6✉]

Multicilia are delicate motile machineries, and how they are accurately assembled is poorly understood. Here, we show that fibrogranular materials (FGMs), large arrays of electron-dense granules specific to multiciliated cells, are essential for their ultrastructural fidelity. Pcm1 forms the granular units that further network into widespread FGMs, which are abundant in spherical FGM cores. FGM cores selectively concentrate multiple important centriole-related proteins as clients, including Cep131 that specifically decorates a foot region of ciliary central pair (CP) microtubules. FGMs also tightly contact deuterosome-procentriole complexes. Disruption of FGMs in mouse cells undergoing multiciliogenesis by Pcm1 RNAi markedly deregulates centriolar targeting of FGM clients, elongates CP-foot, and alters deuterosome size, number, and distribution. Although the multicilia are produced in correct numbers, they display abnormal ultrastructure and motility. Our results suggest that FGMs organize deuterosomes and centriole-related proteins to facilitate the faithful assembly of basal bodies and multiciliary axonemes.

[1] State Key Laboratory of Cell Biology, Shanghai Institute of Biochemistry and Cell Biology, Center for Excellence in Molecular Cell Science, Chinese Academy of Sciences, Shanghai 200031, China. [2] University of Chinese Academy of Sciences, Beijing 100049, China. [3] School of Life Science and Technology, ShanghaiTech University, Shanghai 201210, China. [4] Tsinghua-Peking Joint Center for Life Sciences, School of Life Sciences, Tsinghua University, Beijing 100084, China. [5] Institute of Biomedical Sciences, College of Life Sciences, Key Laboratory of Animal Resistance Biology of Shandong Province, Collaborative Innovation Center of Cell Biology in Universities of Shandong, Shandong Normal University, Jinan 250014, China. [6] School of Life Science, Hangzhou Institute for Advanced Study, University of Chinese Academy of Sciences, Hangzhou 310024, China. [7] These authors contributed equally: Huijie Zhao, Qingxia Chen. ✉email: yanx@sibcb.ac.cn; xlzhu@sibcb.ac.cn

Mammalian multicilia beat back-and-forth to generate luminal fluids in epithelia such as the trachea and brain ventricles. Their formation requires stepwise biogenesis of hundreds of centrioles from parental centrioles and numerous deuterosomes to serve as basal bodies, followed by accurate construction of intricate axonemes consisting of the "9 + 2" microtubule arrays, radial spokes, and dynein arms. Differentiating multiciliated cells (MCCs) accordingly synthesize hundreds of the protein components in large quantities[1–8]. How the MCCs accomplish this task in a highly ordered and faithful manner is poorly understood. We suspected that there might be global organizers in the cells to arrange the centriole-assembly platforms and bulk protein components and promote their coordination, instead of leaving them unattended to function spontaneously.

Fibrogranular materials (FGMs) are ideal candidates for the suspected organizers. FGMs are initially identified through electron microscopy (EM) as widespread electron-dense arrays of 40–80 nm osmophilic fibrous granules (FGs) specifically in MCCs undergoing the centriole amplification. Firstly, FGMs frequently emerge adjacently to or even enclose deuterosomes. Early proposals suggest their functions as deuterosome precursors[9–11]. Mammalian deuterosome components Deup1, Cep152, and Plk4, however, do not display FGM-like localizations[8,12]. Secondly, their only component identified to date, Pcm1[13], is also a major structural protein of the centriolar satellites (CSs), pericentrosomal granules of 70–100 nm in non-MCCs that contain many centriole-related proteins and may regulate their turnover rates and centriolar targeting[14–19]. Thirdly, a cytoplasmic form of E2f4 has been shown to reside in FGMs to control the deuterosome formation and centriole amplification[20]. Finally, FGMs are recently proposed to mediate centriole amplification independently of deuterosomes and parental centrioles[21,22].

In this study, we show that FGMs tightly associate with deuterosomes and selectively concentrate Cep135, Cep120, Cep131, Cep215, Ofd1, Pericentrin (Pcnt), and Rootletin as clients. Pcm1 is an essential structural protein for FGMs. Its depletion markedly alters deuterosome size, number, and distribution and results in deregulated targeting of client proteins to nascent centrioles or cilia. Although multicilia numbers are not affected, they display severe ultrastructural defects and abnormal motility.

## Results

### FGMs are widespread FG networks containing large sponge-like core structures.
In cultured mouse tracheal epithelial cells (mTECs) undergoing MCC differentiation, the FGM component Pcm1 was upregulated following time and distributed as punctate networks interspaced with large (up to a few micrometers in diameter) porous condensates, that mingled with deuterosomes and parental centrioles in early stages (II–IV) of the centriole amplification (Fig. 1a–c)[8]. In late stages (V–VI), the number of the condensates decreased but their size increased (Fig. 1b–e). In stage VI, the punctate Pcm1 networks became enriched around the proximal end of the polarized basal bodies (Fig. 1b, c). Immuno-EM confirmed that our anti-Pcm1 antibody recognized the osmophilic FGs[13] that were either scattered in the cytoplasm or arrayed in FGMs (Fig. 1f). Tightly packed FGM cores (Fig. 1f, circles), corresponding to the condensates in the SIM images (Fig. 1b, c), emerged in loosely packed FGMs (Fig. 1f, arrows), corresponding to the dispersed punctate networks (Fig. 1b, c).

Focused ion beam scanning EM (FIB-SEM) revealed that FGM cores appeared as piles of tightly-packed FG arrays in serial z-sections and spherical FG aggregates in 3D-reconstructed EM images (Fig. 1g and Supplementary Movies 1, 2). 3D modeling revealed that the FGs interconnected into sponge-like porous architectures that were filled with less electron-dense surrounding materials (Fig. 1h and Supplementary Movie 2).

In mouse ependymal cells (mEPCs) undergoing the centriole amplification, Pcm1 condensates were observed both in the cytoplasm and on the nuclear surface, to which deuterosomes have been reported to associate[23]. They were usually smaller in size and less discrete than those in mTECs (Supplementary Fig. 1a vs. Fig. 1b). In addition, punctate Pcm1 networks were observed to highly concentrate around the parental centrioles (Supplementary Fig. 1a). The nuclear association was peculiar to mEPCs because it was not observed in mTECs (Supplementary Fig. 1b). Together, these results imply that FGM networks could store centriolar and ciliary components in their sponge-like cores and host deuterosomes to function as the speculated organizers.

**Pcm1 is an essential structural component of FGs.** As cultured mEPCs are highly efficient for siRNA-mediated protein depletion[24,25], we depleted Pcm1 in them (Fig. 2a–c) to test whether it could be an essential FG protein for the FGM formation. Interestingly, deuterosomes still formed massively and produced procentrioles in the Pcm1-depleted cells (Fig. 2c–f and Supplementary Fig. 2). Transmission EM analyses on ultrathin sections containing deuterosomes (Fig. 2g) indicated that typical FGMs were observed in 21 out of 22 sections of control mEPCs, whereas no recognizable FGMs or FGs were scored in all the 17 sections of mEPCs treated with Pcm-i1.

To exclude possible off-target effects of RNAi, we created adenovirus capable of expressing an siRNA-insensitive Pcm1 and confirmed its correct expression as a GFP fusion protein by immunoblotting (Fig. 2h). In the Pcm1 siRNA-treated mEPCs, the GFP-Pcm1 displayed typical FGM patterns in the cytoplasm and on the nuclear surface (Fig. 2i), indicating reformation of the FGM networks. Therefore, both FGs and FGMs require Pcm1.

**Pcm1 depletion alters deuterosome number, size, and distribution.** We then systematically analyzed effects of the FGM disruption. The first phenotype we noticed was that deuterosome numbers increased by >4-fold as compared to the control cells (Fig. 2c, d and Supplementary Fig. 2a, b). Secondly, deuterosome sizes were generally reduced (Fig. 2c, e and Supplementary Fig. 2a, c). In the control mEPCs, deuterosome sizes tended to elevate from stage II and maximized in stage IV, accompanied by increased numbers of their associated procentrioles (Fig. 2c, f and Supplementary Fig. 1)[8,26]. In the Pcm1-depleted mEPCs, however, deuterosomes in even stage-IV cells were still small and each only supported the biogenesis of a few procentrioles (Fig. 2c, e, f), making it sometimes difficult to discriminate the precise stages. The average deuterosome size in the Pcm1-depleted mEPCs (Fig. 2e and Supplementary Fig. 2c) was only equivalent to that in intact early stage-II mEPCs[26], suggesting a failure in deuterosome growth. Thirdly, deuterosomes became widely spread in the Pcm1-depleted mEPCs. The crop size that we used was usually sufficient to cover all the deuterosomes of a control mEPC, but frequently unable to do so for Pcm1-depleted mEPCs (Fig. 2c and Supplementary Fig 2; also see Fig. 6 and Supplementary Fig. 5), suggesting that FGMs also constrain deuterosome distribution. Consistently, deuterosomes became co-distributed with the FGM networks generated by GFP-Pcm1 in the rescue experiments (Fig. 2i). GFP-Pcm1 also significantly restored the number and size of deuterosomes as compared to Centrin1-GFP (Fig. 2j, k).

When Pcm1 was depleted in mTECs using an adenovirus expressing an shRNA (shPcm-i1) identical to Pcm-i1 in the targeting sequence, we also observed increased deuterosome numbers and a mild decrease (4.4% on average) in deuterosome size in Pcm1-depleted cells as compared to mTECs

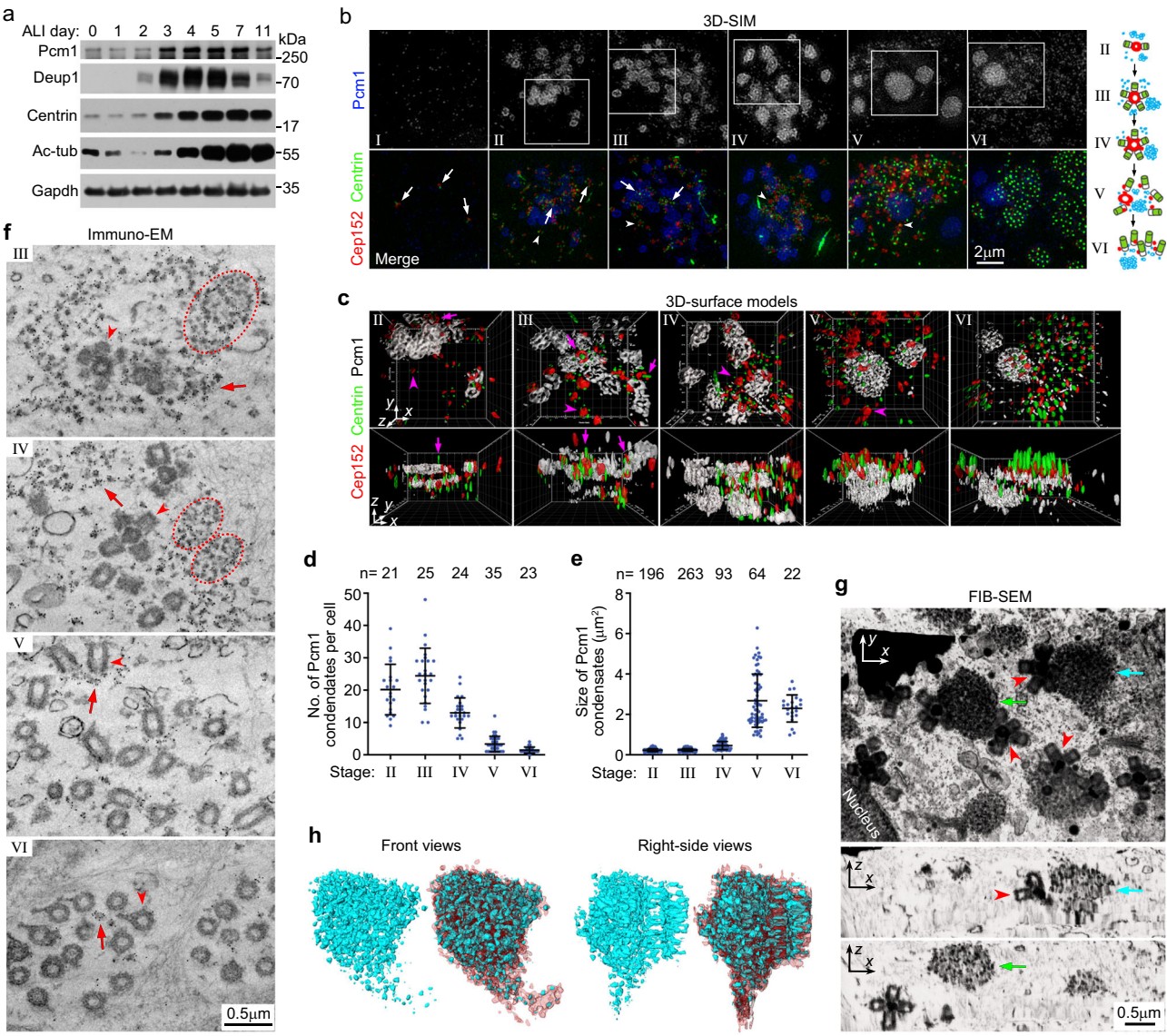

**Fig. 1 Pcm1 decorates loose FGMs and tight FGM cores in mTECs undergoing centriole amplification. a** Expression pattern of Pcm1 during mTEC differentiation. mTECs were induced to differentiate into MCCs by culturing at an air-liquid interface (ALI). Total cells collected at the indicated days were used for immunoblotting. Deup1, Centrin, and acetylated tubulin (Ac-tub) were used to indicate the abundance of deuterosomes, centrioles, and cilia, respectively[8]. Gapdh served as loading control. Two biologically independent experiments were performed and images from the same experiment were presented. **b**, **c** Subcellular distribution of Pcm1 in different stages (I–VI) of the centriole amplification. Presented are typical z-projected 3D-SIM (three dimensional structured illumination microscopy) images of mTECs at day 3 (**b**) and the top and side views of 3D reconstructions (**c**) for the framed regions. Cep152 decorated parental centrioles (arrows), deuterosomes (arrowheads indicating typical ones), and, in stages V–VI, a spot at the proximal side of the nascent centrioles or basal bodies in mTECs[8]. Centrin served as a centriole marker. Diagrams in **b** illustrate typical localization patterns of the proteins during the indicated stages of the deuterosome-mediated centriole biogenesis defined previously[8]. Briefly, deuterosomes start to form in stage II and are enlarged in stage III; the deuterosomal Cep152 displays radial appendages in stage IV; nascent centrioles are released from deuterosomes in stage V, followed by their apical polarization, clustering, and ciliogenesis in stage VI. **d**, **e** Quantifications on the number (**d**) and size (**e**) of the Pcm1 condensates. At least 21 cells from two independent experiments were scored. Both sample dots and mean ± s.d. are presented. **f** Immuno-electron microscopy (Immuno-EM) confirmed the presence of Pcm1 in loose FGMs (arrows) and FGM cores (circles). mTECs at ALI day 3 were immuno-labeled with 10-nm gold particles. Arrowheads indicate typical deuterosome-procentriole complexes (stages III and IV) or basal bodies (stages V and VI). Immuno-EM experiments were performed once. **g**, **h** FGM cores appeared as porous condensates of interconnected FGs. Serial z-stack mTEC images acquired by FIB-SEM (Focused Ion Beam Scanning Electron Microscope) were 3D reconstructed (Supplementary Movie 1) and presented as the top view and two digital z-sections at the arrow-indicated positions (**g**). The arrowheads indicate deuterosome-procentriole complexes spatially adjacent to FGM cores. The structural information for the FGM core pointed by the blue arrow (**g**) was segmented for a 3D view (Supplementary Movie 2) and present as side views (**h**). The high electron-dense FGs were rendered blue and their surrounding less electron-dense materials red. Note that the bottom region contains loosely packed FGMs, possibly in the process of assembly into or disassembly from the core. Two biologically independent experiments were performed. Source data are provided in the Source data file.

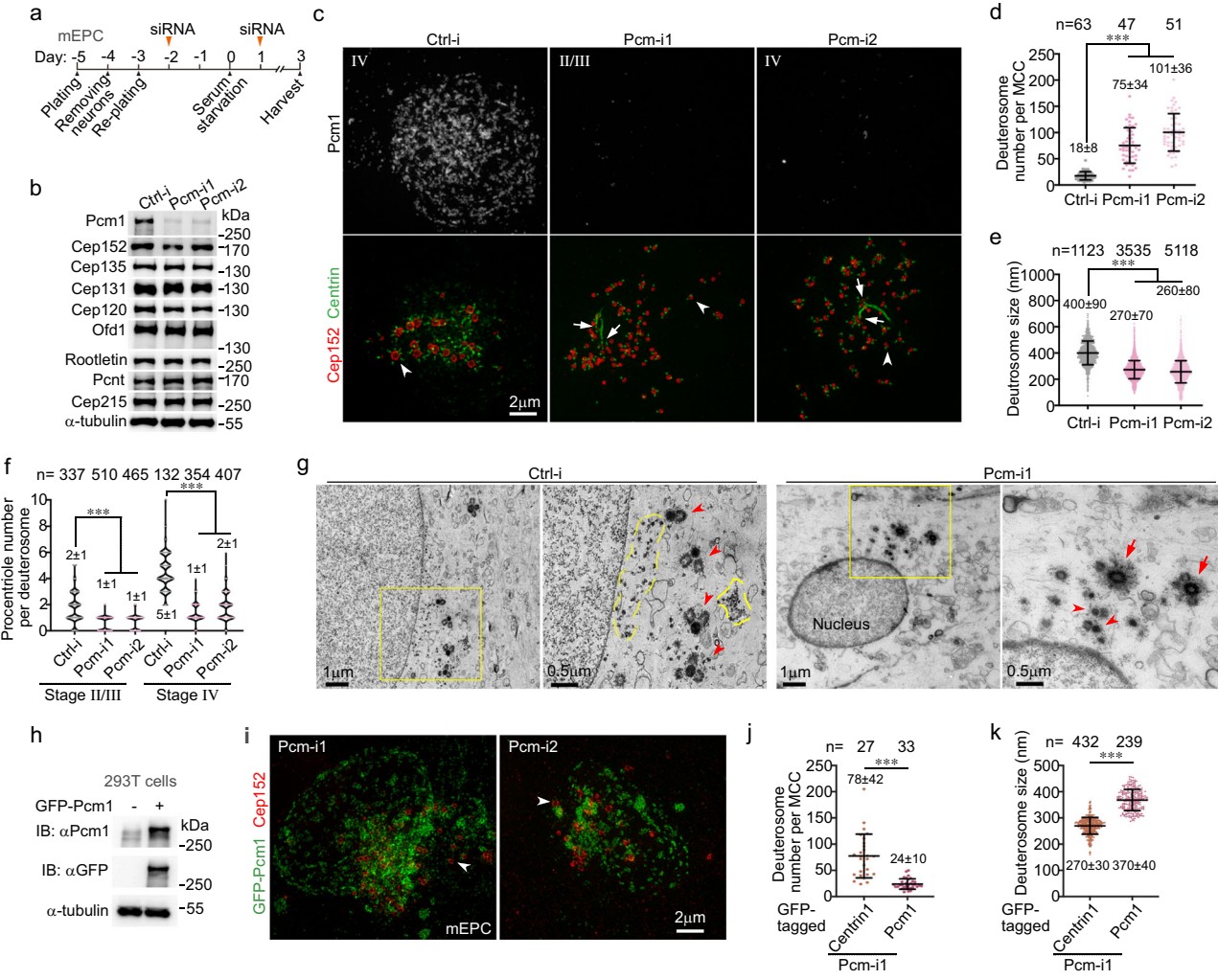

**Fig. 2 Depletion of Pcm1 abolishes FGMs and affects deuterosome growth and number.** Arrows and arrowheads respectively indicate parental centrioles and representative deuterosomes. Both sample dots and mean ± s.d. are presented in quantification results. Two-tailed Mann–Whitney $U$-test: ***$P <$ 0.001. **a** Experimental scheme for Pcm1 RNAi. Mouse ependymal cell (mEPC) precursors were cultured from P0 mouse brain tissues at day −5 and induced to differentiate into MCCs through serum starvation at day 0. The cells were transfected with siRNAs twice and collected at day 3 for analyses. **b** Two independent siRNAs (Pcm-i1 and Pcm-i2) effectively depleted Pcm1 without affecting the indicated Centriole-related proteins, which are related to the current study. α-tubulin served as loading control. Ctrl-i control siRNA. **c** 3D-SIM images of representative mEPCs. Cep152 decorated both parental centrioles and deuterosomes. Centrin served as a centriole marker. **d–f** Depletion of Pcm1 increased deuterosome number and decreased deuterosome size. The number (**d**) and size (**e**) of deuterosomes and the number of deuterosome-associated procentrioles (**f**) were scored from at least 47 cells in stages II–IV from three independent experiments. Cep152 served as deuterosome marker. Also see Supplementary Fig. 2, in which Deup1 was used as deuterosome marker. **g** Electron micrographs showing that the depletion of Pcm1 abolished FGMs. FGMs are outlined with dashed lines. Two biologically independent experiments were performed and images from the same experiment were presented. **h** Correct expression of siRNA-insensitive GFP-Pcm1 in HEK293T cells. Immunoblotting (IB) was performed using antibodies to Pcm1 and GFP, respectively. **i** GFP-Pcm1 rescued FGM formation in Pcm1-depleted mEPCs. mEPCs were infected with adenovirus to express an RNAi-insensitive GFP-Pcm1 in addition to the siRNA treatment (**a**). **j**, **k** GFP-Pcm1 rescued the Pcm1 RNAi-induced alterations in deuterosome number (**j**) and size (**k**). Cep152 served as deuterosome marker. Centrin1-GFP, expressed also through adenoviral infection at day −1, served as negative control. At least 27 mEPCs in stages II–IV from two independent experiments were scored. Two biologically independent experiments were performed. Source data are provided in the Source data file.

expressing a control shRNA (shCtrl-i) (Supplementary Fig. 3a–c), suggesting conservation of the Pcm1 functions in multiciliated mTECs.

**Pcm1 depletion impairs ciliary motility by causing ultra-structural defects in basal bodies and axonemes.** Next we investigated whether FGMs affected multicilia function. Similar to the previous study in mTECs[6], the depletion of Pcm1 in mEPCs or mTECs did not markedly affect the production of basal bodies (Fig. 3a–d and Supplementary Fig. 3d, e). Ciliary motilities, however, were markedly altered by the Pcm1 depletion. Multicilia

in the majority (>85%) of the control mEPCs or mTECs beat rapidly in the expected back-and-forth (planar) manner[1,24,27]. By contrast, multicilia with abnormal motility, including immotility and rotatory beat, were observed in most (>74%) of the Pcm1-depleted MCCs (Fig. 3e–g and Supplementary Fig. 3f, g and Movies 3, 4). In rescue experiments, the planar beat was observed in ≥86% of the Pcm-i1-treated mEPCs expressing GFP-Pcm1 but only in ≤19% of the cells expressing Centrin1-GFP (Fig. 3h–j and Supplementary Movie 5), indicating a specific rescue of the ciliary motility defects by the exogenous Pcm1. Thus, Pcm1 is critical for proper ciliary motility.

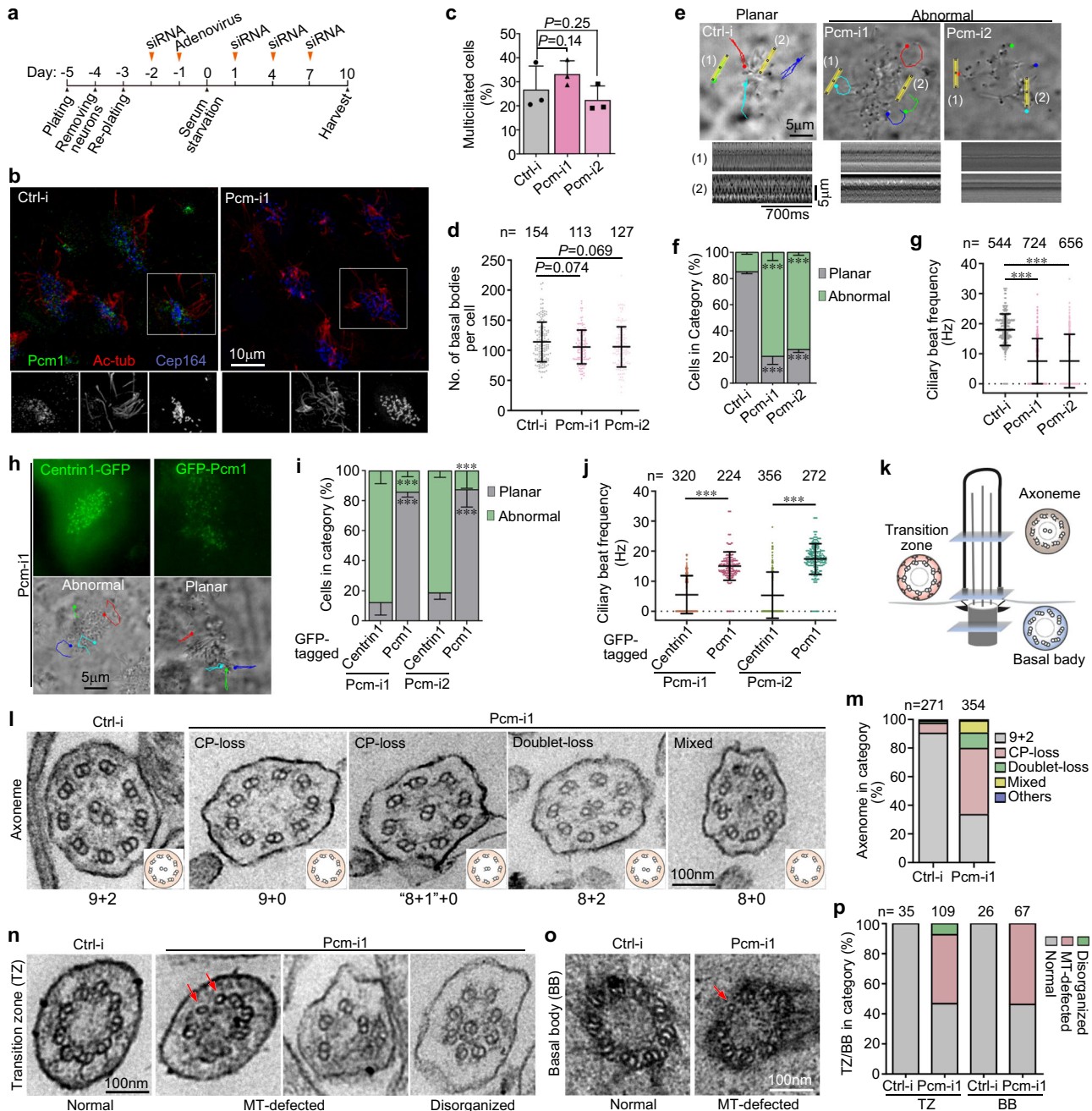

EM studies revealed that the Pcm1 depletion resulted in severe ultrastructural defects in both basal bodies and axonemes. The typical "9 + 2" microtubule arrangement was observed in 91% of axonemal cross-sections from the control mEPCs (Fig. 3k–m). The value was only 34% for the Pcm-i1-treated cells and the remaining 66% contained various abnormal axonemal structures, including loss of the central pair microtubules (CP-loss; 46%) or peripheral doublets (doublet-loss; 11%) or both (mixed; 8%) (Fig. 3l, m). In addition, 53% of ciliary transition zones and 54% of the basal bodies in the Pcm-i1-treated cells also displayed defects in microtubule organization, comparing to the 100% normal ultrastructure in the control cells (Fig. 3n–p). Therefore, Pcm1 is critical for the assembly of proper basal body and ciliary structures. As motile cilia lacking the CP are usually rotational[24,28], these EM results are also consistent with the motility assays (see Fig. 3e).

**FGMs enrich multiple centriole-related proteins**. As Pcm1 itself does not localize in the deuterosome, basal body, or cilium in MCCs (Fig. 1)[6,13], we attributed the Pcm1 depletion-induced changes in these organelles (Figs. 2 and 3) to the disruption of FGMs (Fig. 2). We thus used an ascorbic acid peroxidase (APEX)-mediated proximity labeling approach[29,30] to investigate whether FGMs enriched centriolar or ciliary components as client proteins. As our adenovirus for exogenous Pcm1 expression did not efficiently infect mEPCs possibly due to Pcm1's large cDNA size, we identified another FGM protein, Cep131/Azi1, a CS component that interacts with Pcm1[18,31,32], to serve as a bait. Cep131 co-distributed nicely with Pcm1 and appeared to be fillings of the porous FGM cores (Fig. 4a). In mEPCs expressing APEX2-Cep131, biotinylated proteins were concentrated in FGMs, especially FGM condensates (Fig. 4b).

**Fig. 3 Pcm1 depletion leads to defected basal bodies and axonemes and abnormal ciliary motility.** Pooled data are presented as mean ± s.d. with (dot plots) or without (histograms) sample dots. Unpaired two-tailed student's *t*-test (**c**, **f**, **i**) or Mann–Whitney *U*-test (**d**, **g**, **j**): ***P < 0.001. **a** Experimental scheme. mEPCs treated as shown were collected at day 10 for analyses **b–l** Adenoviral infections were performed only in rescue experiments (**h**) to express siRNA-insensitive GFP-Pcm1 and Centrin1-GFP (negative control). **b–d** Pcm1 was not required for MCC differentiation or basal body production. Ac-tub and Cep164 labeled cilia and basal bodies, respectively (**b**). Percentages of MCCs (**c**) were quantified from three independent experiments. At least 300 cells were scored in each experiment and condition. Basal body numbers (**d**) were quantified from at least 113 MCCs pooled from three independent experiments. **e–g** Depletion of Pcm1 altered ciliary motility. Ciliary motions of living mEPCs were captured at 7-ms intervals by using high-speed differential interference contrast microscopy. The micrographs overlaid with the trajectories of four recognizable cilia (**e**) were cropped from the frames at 56 ms in Supplementary Movie 3. Kymographs of a single cilium (1) and multiple cilia (2) are presented. Quantification results (**f**, **g**) were from three independent experiments. MCCs with their cilia beating predominantly in a back-and-forth pattern were scored as "planar", whereas those predominantly containing immotile and/or irregularly-beating cilia were scored as "abnormal" (**f**). At least 43 MCCs were analyzed in each experiment and condition. Ciliary beat frequencies (**g**) were quantified from traceable individual cilia. **h–j** Expression of RNAi-insensitive GFP-Pcm1, but not Centrin1-GFP, rescued the Pcm1 depletion-induced abnormal ciliary motility. The GFP and bright-field micrographs (**h**) were cropped respectively from the first and the corresponding 56-ms frames of Supplementary Movie 5. Quantification results (**i**, **j**) were from three independent experiments. At least 18 GFP-positive MCCs were analyzed in each experiment and condition (**i**). Ciliary beat frequencies (**j**) were quantified from traceable individual cilia. **k** Diagrams for the location and typical ultrastructure of the basal body, transition zone, or axoneme. **l–p** Depletion of Pcm1 resulted in abnormal axonemes (**l**, **m**), transition zones (**n**, **p**), or basal bodies (**o**, **p**). Two biologically independent experiments were performed. Quantification results (**m**, **p**) were from cross-sections collected from two independent experiments. Only micrographs allowing clear judgment on the ultrastructure were scored. Axonemes with an extra doublet or irregular doublet arrangement were classified as "others". Transition zones containing singlet (arrows) and/or lacking doublet microtubules (MT) were scored as "MT-defected", whereas those with an abnormal arrangement of the nine doublets were scored as "disorganized". TZ transition zone, BB basal body, CP central pair of microtubules, MT microtubule. Source data are provided in the Source data file.

We reasoned that FGM proteins would be preferentially labeled in FGM-containing but not FGM-free mEPCs, and thus analyzed proteins differentially proximity-labeled in Pcm1-depleted and control mEPCs. Shotgun mass spectrometry revealed that Cep131 was the topmost hit in both samples, whereas Pcm1 was highly enriched in the control sample (Fig. 4c), suggesting successful RNAi and proximity labeling. Five hundred and twenty-six proteins were enriched by at least 2-fold in the Ctrl-i-treated sample as compared to the Pcm-i1-treated sample (Supplementary Data 1). 16 and 8% of them were related to the centriole and cilium, respectively (Fig. 4c). We examined several top hits of centriole-related proteins in mTECs based on antibody availability and found that Cep135, Cep120, Ofd1, Rootletin, Pcnt, and Cep215 displayed FGM localizations (Fig. 4c, d). Similar to Cep131 (Fig. 4a), Cep135, Cep120, Ofd1, and Rootletin were highly concentrated in the Pcm1 condensates (Fig. 4d). Pcnt and Cep215, however, displayed punctate localizations that overlapped mainly with the punctate Pcm1 networks (Fig. 4d). Immuno-EM confirmed that the immuno-gold particles for Cep135 and Rootletin were prominent in the FGM cores, whereas those for Cep215 and Pcnt tended to label dispersed FGMs (Fig. 4e). Immunoblotting indicated that these proteins displayed different extents of upregulation in the differentiating mTECs (Fig. 4f).

We also performed a brief survey on additional candidate proteins. As all our identified FGM proteins except for Rootletin were CS components[14,15], we examined additional CS proteins[15], some of which (Cep290, Odf2, and Centrobin) were also candidate FGM proteins (Supplementary Data 1). We also examined two ciliary proteins in our list, Ift81 and Bbs5 (Supplementary Data 1)[33,34]. Of these proteins, only Cep290, a ciliary transition zone protein[35], displayed FGM-like distributions (Supplementary Fig. 4). Therefore, FGMs only enrich a subset of CS proteins. Furthermore, additional FGM proteins need to be validated individually.

**Newly identified FGM proteins target to nascent centrioles in diverse stages**. We investigated when and where the identified FGM proteins were targeted to nascent centrioles or cilia in differentiating mTECs. As some FGMs tended to accumulate around parental centrioles (Fig. 5a–f, arrows), in stages II–IV we only examined procentrioles associated with deuterosomes. In cycling cells, Cep135 and Cep120 are located to the centriolar cartwheel and proximal wall, respectively, and critical for proper centriole elongation[36–38], whereas Ofd1 localizes to centriolar distal end to support the distal appendage assembly[39]. In stage-I mTECs, these three proteins displayed similar centriolar localizations as in the cycling cells (Fig. 5a–c). On procentrioles, Cep135 was hardly detected till stage IV, whereas Cep120 and Ofd1 were readily observed from stage III (Fig. 5a–c). Rootletin is a component of the ciliary rootlet and the filamentous networks connecting parental centrioles[40,41]. It distributed as bundles radiated from parental centrioles in stage I and was incorporated into the nascent rootlets from stage V (Fig. 5d). Pcnt and Cep215 are interacting partners located in the pericentriolar material in cycling cells[42,43]. They localized at the proximal side of parental centrioles in stage I and became abundant around the proximal wall of nascent centrioles from stage IV (Fig. 5e, f). Interestingly, they also displayed weak deuterosome localizations in early stages (Fig. 5e, f).

Interestingly, Cep131 specifically localized to the top region of the centriolar Centrin streak in stage VI (Fig. 5g). We have previously observed that the Centrin streak in stage-VI mEPCs appears to extend into the ciliary shaft[26]. Further examinations revealed that the Centrin streak in stage-VI mEPCs usually had a lateral constriction (Fig. 5h, i). The commonly-known centriolar Centrin was actually the region below the constriction and was relatively fixed in length, whereas the top region was highly variable and could exceed the bottom region by several folds. Cep131 specifically localized to this variable top region (Fig. 5h). Immunostaining of Cep290 (Fig. 5i)[35] suggested that the transition zone starts from the constriction. The top region of Centrin streaks was mostly buried in the transition zone when it was short but apparently extended into the ciliary shaft when it was long (Fig. 5i). Therefore, Cep131 and, partly, Centrin mark a previously uncharacterized structure that extends from the top of the basal body into the cilium, which we term as "CP-foot" (see below).

**Pcm1 depletion deregulates the centriolar and ciliary targeting of FGM client proteins**. As the FGM protein levels in mEPCs were not altered upon Pcm1 RNAi (Fig. 2b), we investigated their subcellular localizations. We observed that their FGM localizations were abolished (Fig. 6a–c and Supplementary Fig. 5a–d).

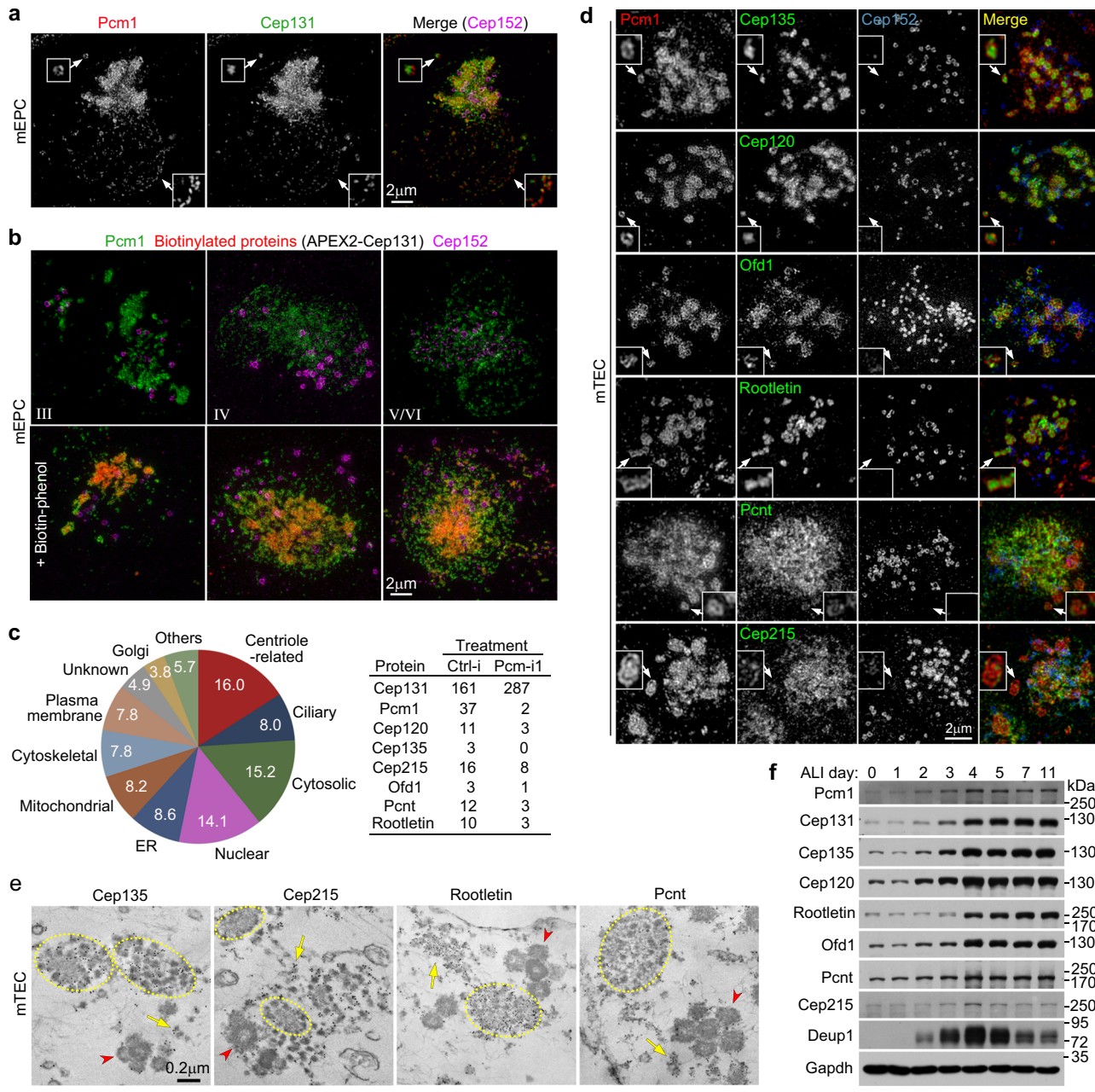

**Fig. 4 Proximity labeling through APEX2-Cep131 identifies multiple FGM components. a** Cep131 colocalized with Pcm1 in FGMs. mEPCs fixed at day 3 postserum starvation were immunostained and imaged by 3D-SIM. Cep152-labeled deuterosomes served as MCC marker. The magnified insets show details for cytoplasmic (top) and perinuclear (bottom) FGMs. **b** APEX2-Cep131 induced biotinylation of FGM proteins. mEPCs infected with adenovirus at day −1 to express APEX2-Cep131 were either mock-treated (upper panels) or treated with biotin-phenol and $H_2O_2$ to trigger the proximity labeling reaction (lower panels) at day 3, followed by fluorescent staining and 3D-SIM. Biotinylated proteins were visualized with Alexa Fluor 546-labeled streptavidin[30]. **c** FGM candidate proteins identified by differential mass spectrometric analysis. APEX2-Cep131-expressing mEPCs were transfected respectively with Ctrl-i and Pcm1-i1, followed by the proximity labeling with biotin-phenol. The biotinylated proteins were precipitated using streptavidin resin and subjected to shotgun mass spectrometry. The pie chart indicates protein compositions according to subcellular localizations. The table lists peptide spectrum matches (PSM) of Pcm1 and seven newly identified FGM proteins in the control (Ctrl-i) and Pcm1-depleted mEPC samples. **d** Subcellular localization of the indicated FGM components with Pcm1 in mTECs at day 3. Insets are magnified images of typical FGM cores. **e** Immuno-EM confirmed FGM localization of the indicated proteins in mTECs at day 3. Cep135 and Cep215 were immune-labeled with 10-nm gold particles, whereas Rootletin and Pcnt by 5-nm gold particles. Arrows indicate loose FGMs. FGM cores are encircled. Arrowheads point to typical procentriole-deuterosome complexes. Immuno-EM experiments were performed once. **f** Expression pattern of the FGM proteins. mTECs collected at the indicated ALI days were used for immunoblotting. Deup1 served as deuterosome marker. Gapdh served as loading control. Two biologically independent experiments were performed and images from the same experiment were presented. Source data are provided in the Source data file.

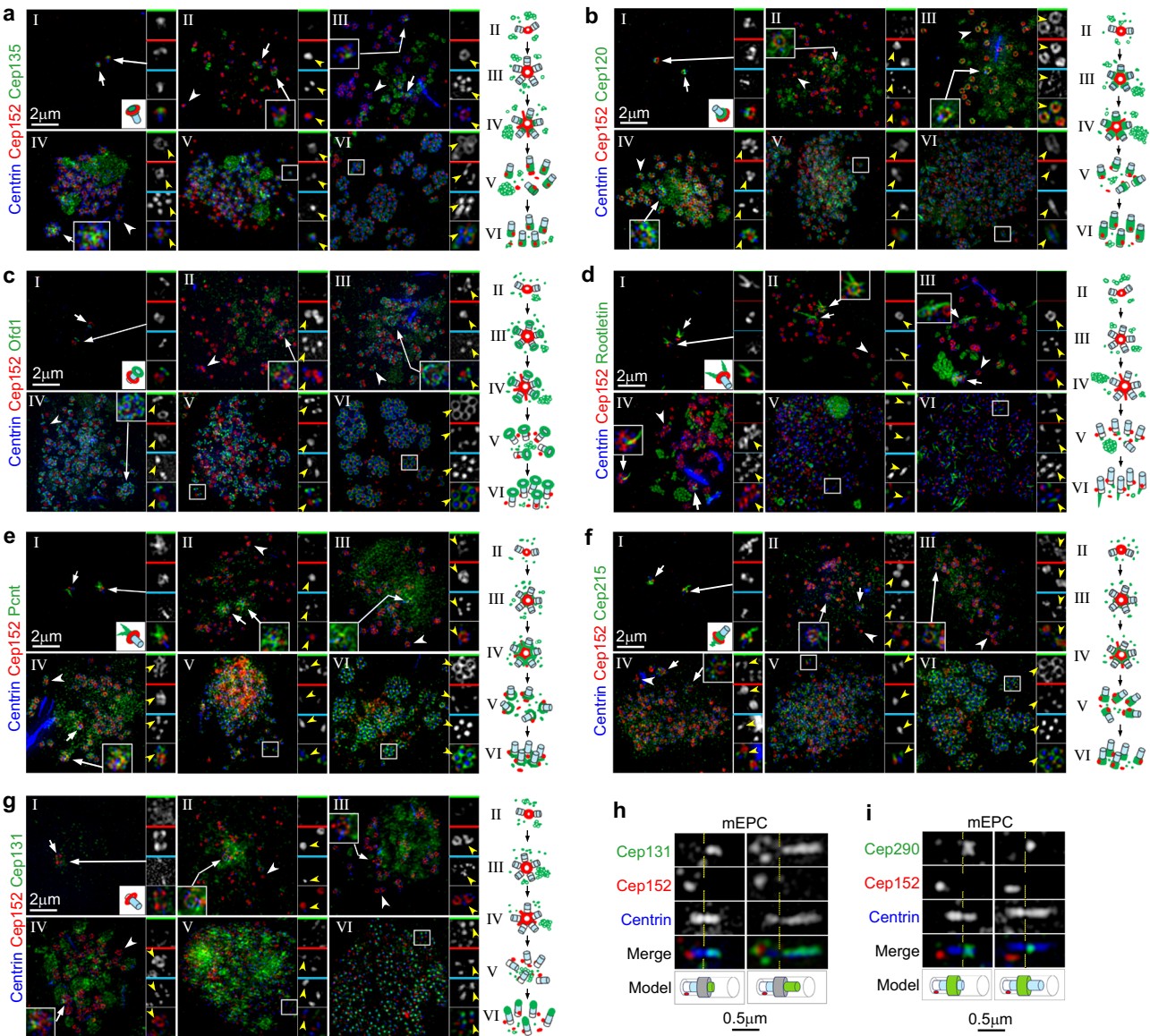

**Fig. 5 Newly identified FGM proteins localize to nascent centrioles and cilia in diverse centriole amplification stages. a–g** Representative 3D-SIM images for the indicated FGM proteins (green) in mTECs fixed at day 3. White arrows indicate parental centrioles, one of which was magnified. White arrowheads and frames respectively indicate typical regions containing deuterosomes and basal bodies that were magnified to show details. Yellow arrowheads in the color-coded insets point to immunofluorescent signals related to typical nascent centrioles. The diagrams illustrate FGM protein localizations in stage-I parental centrioles and during the deuterosome-mediated centriole biogenesis (II–VI). **h, i** Cep131 localized to a novel Centrin-positive structure protruded from the center of transition zone into cilia. Shown are typical basal bodies in stage-VI mEPCs at day 3. Cep152 labeled the centriolar proximal side. Cep290 served as transition zone marker. The constrictions of Centrin streaks are marked by dashed lines. Note that Cep131 localized to the region of Centrin streaks above the constriction and sometimes also displayed additional punctate immunofluorescent signals around the basal bodies (**h**).

Only Rootletin tended to polymerize into numerous filaments in mEPCs regardless of Pcm1 (Supplementary Fig. 5e). The centriolar localizations of Cep135, Cep120, and Ofd1 were weak or undetectable in stage II and increased in later stages in the control mEPCs, but became strong in all the Pcm1-depleted cells undergoing the centriole amplification (Fig. 6a–d and Supplementary Fig. 5a, b), indicating excess or premature centriolar targeting of the proteins in the absence of FGMs. Cep215 and Pcnt displayed obvious deuterosome localizations outside that of Cep152 in stages II and III, though the Pcm1 depletion did not obviously change their deuterosomal and centriolar localizations (Supplementary Fig. 5c, d).

Interestingly, the ciliary Cep131 and Centrin double-positive streaks in stage VI (Fig. 5g–i) were markedly elongated upon the

Pcm1 depletion (Fig. 6b). To quantify the difference, we measured the length of the streaks at day 5 as 0.6 ± 0.2 μm for the control cells and 1.2 ± 0.3 μm for the Pcm1-depleted cells (Fig. 6e, f). When transition zone marker Cep162[34,44] was used to discriminate the ciliary Centrin from the centriolar Centrin, we noticed that the ciliary Centrin streaks in the Pcm1-depleted mEPCs grew even longer at day 10 and some reached several micrometers (Fig. 6g). Co-staining with acetylated tubulin indicated that the streak was positioned in the axonemal central lumen (Fig. 6g). Furthermore, the CP marker Hydin[2] distributed right on top of the Centrin streak regardless of its length (Fig. 6h), suggesting that this Centrin and Cep131 double-positive streak serves as "CP-foot", i.e., the base of CP microtubules. Notably, many CP-feet in the Pcm1-depleted cells lacked the Hydin

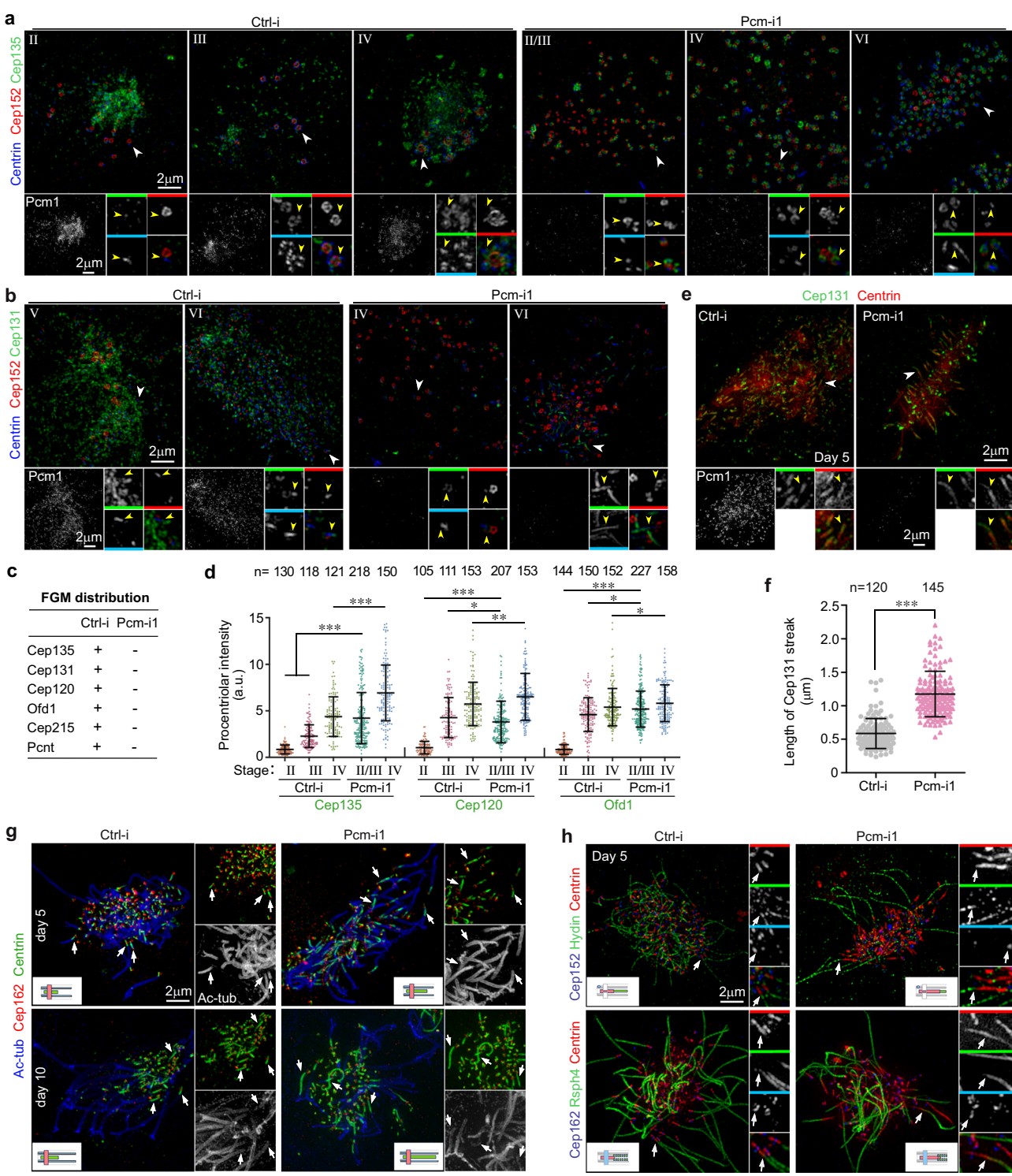

staining (Fig. 6h), echoing the lack of CP microtubules (Fig. 3l, m). Immunostaining for Rsph4 indicated that radial spokes partly overlapped with the CP-foot (Fig. 6h). Therefore, the largely elongated CP-feet at day 10 (Fig. 6g, h) would significantly affect the CP and ciliary radial spoke arrays in the Pcm1-depleted mEPCs and contribute to the abnormal ciliary motility (Fig. 3e–g and Supplementary Movie 3).

Similar results were observed in Pcm1-depleted mTECs (Supplementary Fig. 6). Taken together, we conclude that FGMs function to modulate centriolar targeting of the client proteins.

**FGMs are possibly Pcm1 phase separation-induced condensates capable of confining deuterosomes.** Emerging lines of evidence suggest that protein phase separation underlies the formation of dynamic membrane-less organelles[45,46]. To understand how FGMs formed, we labeled FGMs with GFP-Pcm1 and deuterosomes with SNAP-tagged Deup1[26,47] and examined their dynamic behaviors with a spinning disk microscope (Fig. 7a). We observed that deuterosomes colocalized with FGM foci of varying sizes and moved together in the mEPCs ($n = 10$) (Fig. 7b and Supplementary Fig. 7). Sometimes one or several deuterosomes

**Fig. 6 Depletion of Pcm1 causes deregulated centriolar targeting of the identified FGM proteins.** Pooled data are presented as mean ± s.d. with sample dots. Two-tailed Mann–Whitney U-test: *P < 0.05; **P < 0.01; ***P < 0.001. mEPCs treated with siRNAs as in Fig. 3a were fixed at day 3 (**a**, **b**) or day 5 and 10 (**e**, **g**, **h**) and subjected to 3D-SIM. **a**, **b** Representative 3D-SIM images for Cep135 and Cep131. The regions indicated by white arrowheads were magnified to show details. Yellow arrowheads in the color-coded insets point to typical nascent centrioles. The immunofluorescent signals for Centrin were less satisfactory in the four-color immunostaining. Two alternative stages are indicated in cases when we have difficulty to clearly define the precise stage of some cells. **c** A summary for the indicated proteins. See Supplementary Fig. 5 for 3D-SIM images of additional proteins. **d** Procentriolar localizations of Cep135, Cep120, and Ofd1 increased upon the depletion of Pcm1. Procentriolar intensities were quantified from three independent experiments and at least five MCCs in each experiment and condition. Note that, although the Pcm1-depleted cells in stages II and III were indistinguishable, the mixed populations lacked procentrioles with weak centriolar Cep135, Cep120, or Ofd1 that resembled those in the stage-II control cells, suggesting a deregulated procentriolar targeting in stage II. **e**, **f** Depletion of Pcm1 increased the length of the Cep131 and Centrin double-positive streaks in multiciliated mEPCs. The quantification results (**f**) were measured from 3D-SIM images (**e**) collected in two independent experiments. As multicilia are mostly protruded toward the z-axis, length measurement using z-projected images would generate underestimated results. To diminish such underestimation, we only measured five longest Cep131 streaks in each MCC. **g** The CP-foot was positioned in the axonemal central lumen and aberrantly elongated following time in the Pcm1-depleted mEPCs. Ac-tub and Cep162 labeled axonemal microtubules and transition zone, respectively. Arrows indicate representative long CP-feet labeled with Centrin. Diagrams illustrate distribution patterns of the proteins. **h** The CP-foot excluded Hydin and partly overlapped with radial spokes. Hydin is associated with CP microtubules. Rsph4 is a radial spoke subunit. Arrows indicate representative cilia. Source data are provided in the Source data file.

were observed to associate with larger FGM foci (cores), though the resolution of spinning disk microscopy did not allow us to determine whether the deuterosomes were anchored to the surface of these foci or buried inside them. When two cores met, they slowly fused into a larger one and pooled their associated deuterosomes together (Fig. 7b and Supplementary Fig. 7a and Movie 6). The fusion indicates that FGMs possess liquid-like properties[48,49] and explains how an FGM condensates can contain multiple deuterosomes. Fluorescence recovery after photobleaching (FRAP) assays revealed a low extent of recovery (~30%) after photobleaching (Fig. 7c, d), suggesting relatively slow and limited exchanges of GFP-Pcm1 between the condensates and the cytosol.

Secondary structure prediction suggested that Pcm1 is abundant in intrinsically disordered regions (Fig. 7e), which are prone to phase separation[45,46,50]. To verify this, we purified three fragments with high disorder probability (Pcm1N, Pcm1M, and Pcm1C) as His-GFP fusion proteins from E. coli (Fig. 7e, f). Only Pcm1M phase separated into liquid droplets following the addition of PEG as a crowding reagent[51,52], with the droplet size and number positively correlated protein concentrations (Fig. 7g). The droplets were able to fuse rapidly into larger ones in time-lapse imaging (Fig. 7h), confirming their liquid-like properties. Furthermore, FRAP assays revealed highly dynamic protein exchanges between the droplets and the milieu (Fig. 7i, j).

Although Pcm1N and Pcm1C did not form liquid droplets even in the presence of PEG, their protein preparations were clear but sticky at high concentrations. We thus examined whether they could phase separate into hydrogels using a protocol established previously[53]. After centrifugation for 30 min, both proteins formed sticky transparent pellets, which solidified after further incubation (Fig. 7k), indicating their phase separation into hydrogels[46,53]. Therefore, Pcm1 can undergo domain-specific phase separation in vitro.

## Discussion

We demonstrate that FGMs serve as cellular storages to organize some centriolar and ciliary components for their proper assembly. Pcm1 may aggregate into FGs as proposed for CSs[16,18,32], which network into FGMs through intermolecular interactions of Pcm1's disordered regions (Figs. 1, 2, and 7). When intensified, such interactions may result in phase separation and consequently the formation of FGM cores (Figs. 1 and 7)[45,46,50,52,53]. The slow fusion kinetics (within several minutes) and GFP-Pcm1 turnover (Fig. 7b–d and Supplementary Fig. 7a) suggest a viscous liquid-like property of the FGM cores. Such a property echoes the distinct phase separation abilities of different Pcm1 regions into

highly dynamic liquid droplets or relatively static hydrogels (Fig. 7e–j)[45,46,50]. The porous FGM cores then selectively concentrate client proteins such as Cep135, Cep120, Ofd1, and Cep131 to prevent them from premature or excessive assembly into nascent centrioles or cilia (Figs. 1, 4–6 and Supplementary Figs 4–6).

Interestingly, we found that Cep131 is a specific component for a previously unknown ciliary structure, or CP-foot, whose length varied dramatically in mEPCs and mTECs and grew over time during multiciliation (Fig. 6e–h and Supplementary Fig. 6b–d). In protozoan motile cilia or flagella, CP microtubules emerge from the top of basal plate, which lies immediately over the distal transition zone with a thickness of less than 200 nm[54,55]. Disruption of the basal plate in trypanosome leads to CP defects[54]. As the basal plate is not conserved in both morphology and composition even in protozoa[54,55], the CP-foot could be the orthologous structure in vertebrates. While its functions require future clarifications, its marked elongation to up to several micrometers upon the Pcm1 depletion would probably impair both CP and radial spokes and contribute to the ciliary motility and ultrastructural defects (Figs. 3 and 6e–i).

FGMs restrain deuterosomes. Although deuterosomes are known to frequently locate near FGMs (Fig. 1 and Supplementary Fig. 1 and Movie 1)[9–11], the correlation could be due to coincidence. Our live imaging showed that most, if not all, deuterosomes are associated and co-trafficked with FGM foci (Fig. 7b and Supplementary Fig. 7). Accordingly, deuterosomes became dispersed in the absence of FGMs (Figs. 2 and 6 and Supplementary Fig. 3b). The abundant FGMs around the centrosome and on the nuclear surface of mEPCs (Supplementary Fig. 1a) also explain previous observations of the regional enrichments of some deuterosome-procentriole complexes[12,22,23]. In sharp contrast, the FGMs and deuterosomes in mTECs do not associate with the nucleus (Supplementary Fig. 1b). It is thus possible that an mEPC-specific protein or complex tethers FGMs to the nuclear surface (Supplementary Fig. 1a vs. Fig. 1b). As multicilia are dispersed on the entire apical surface in mTECs but confined in a certain apical area (transitional polarization) in mEPCs[1,56], it will be interesting to investigate how FGMs are attached to the nuclear surface and whether the nucleus-associated FGMs control the transitional polarization.

FGMs also control the size and number of deuterosomes. As the depletion of Pcm1 caused overproduction of small deuterosomes in both mEPCs and mTECs (Figs. 2 and 6 and Supplementary Fig. 3b, c), the FGMs-mediated deuterosome confinement might promote the growth of deuterosomes by repressing their overnucleation. Alternatively, fusions of FGM cores containing

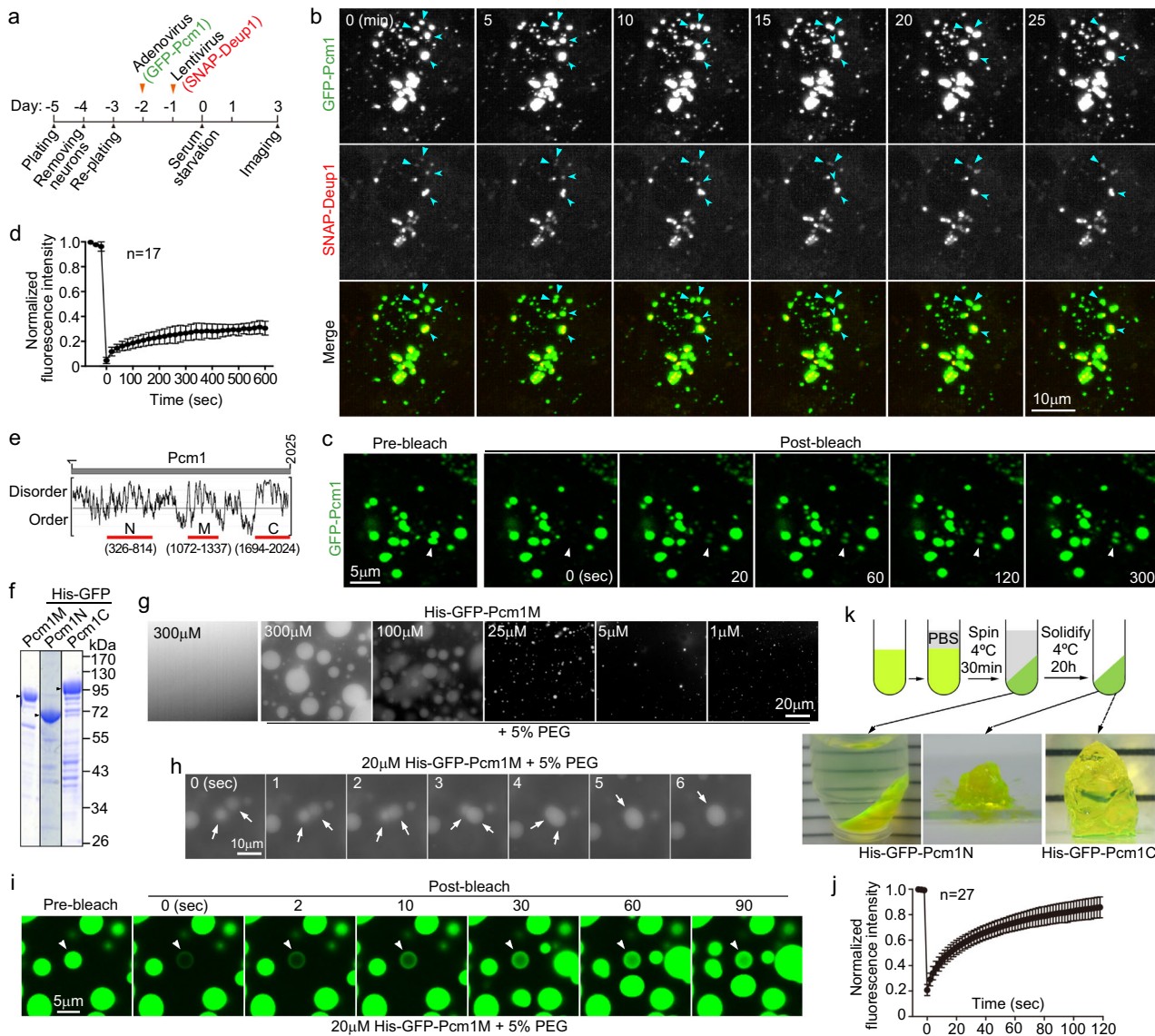

**Fig. 7 FGMs tightly associate with deuterosomes and may form through Pcm1 phase separation. a** Experimental scheme. Wild-type mEPC precursors were infected with adenovirus to express GFP-Pcm1 alone (**c**) or together with lentivirus to co-express SNAP-Deup1 (**b**). SNAP-Deup1 served as deuterosome marker and its expression, controlled by the *Deup1* promoter, indicated cells undergoing the centriole amplification[26]. **b** FGM foci displayed liquid properties and associated with deuterosomes. Living mEPCs positive for GFP-Pcm1 and SNAP-Deup1 were imaged at 5-min intervals using a spinning disk confocal microscope. The image sequences show two fusion events of FGM foci (arrowheads). Note that the deuterosomes were associated and moved with FGM foci. 3D reconstruction was performed for the last frame to show spatial information (Supplementary Movie 6). More examples are presented in Supplementary Fig. 4. Experiment was performed once and multiple cells in different microscopic fields were live imaged. **c, d** FGM condensates displayed slow and limited exchanges of Pcm1 with the cytosol. mEPCs were infected to express GFP-Pcm1 as in **a**. GFP-positive condensates were photobleached, followed by live imaging (**c**). The fluorescence recovery curve (**d**) was from 17 condensates. Two biologically independent experiments were performed. Data are presented as mean ± s.d. **e, f** Secondary structure prediction and expression of Pcm1 deletion mutants. Three fragments of Pcm1 containing predicted disordered regions (**e**) were expressed in *E. coli* as His-GFP-tagged fusion proteins and purified (**f**). **g, h** Pcm1M phase-separated into liquid droplets in the presence of PEG. Purified His-GFP-Pcm1M was diluted to the indicated concentrations with or without 5% PEG3350 and imaged after incubation at 25 °C for 2 min (**g**). The time-lapse images (**h**) show the fusion of two liquid droplets (arrows). Three biologically independent experiments were performed. **i, j** Liquid droplets of Pcm1M rapidly exchanged the protein with the milieu. Liquid droplets prepared as in **h** were photobleached, followed by time-lapse imaging (**i**). The arrowhead points to the representative droplet before and after the photobleaching. The fluorescence recovery curve (**j**) was from 27 droplets. Data are presented as mean ± s.d. **k** Pcm1N and Pcm1C phase-separated into hydrogels. Hundred micromolar of soluble His-GFP-tagged Pcm1N or Pcm1C were topped with PBS and spun for 30 min and photographed against a white striped background to show the transparency of the pellets. After a further incubation for 20 h, the sticky pellets solidified into transparent hydrogels. Source data are provided in the Source data file.

deuterosomes (Fig. 7b and Supplementary Fig. 7) might facilitate deuterosomes to merge into larger ones. The confinement might also be important for the precise assembly of basal bodies and cilia (Figs. 3 and 6) by allowing appropriate accessions to their

components. Cep215 and Pcnt are candidates for the FGM-deuterosome association because of their deuterosome localizations (Supplementary Fig. 5c, d). As Pcnt and Cep215 help to maintain the cohesion between the mother and daughter

centrioles in cycling cells[42,43,57,58], they might also confer the confinement partly by linking nascent centrioles to FGMs. Therefore, our results suggest that the dual organization roles of FGMs on client proteins and deuterosomes enable the faithful construction of basal bodies and axonemes to ensure proper multiciliary motility (Fig. 3 and Supplementary Figs. 3 and 6). Future investigations will identify additional client proteins and elaborate detailed mechanisms.

Despite the increase in deuterosome number, the disruption of FGMs does not significantly impact the eventual basal body number in MCCs (Fig. 3b, d and Supplementary Fig. 3d, e). In Pcm1-depleted mEPCs, this is correlated with the decrease in procentrioles produced per deuterosome (Fig. 2c–f). Interestingly, it has recently been reported that MCCs lacking parental centrioles, deuterosomes, or even both are able to keep a relatively constant centriole number as well[21,59]. How MCCs manage to maintain the robustness of the centriole amplification is thus an important question to be clarified in the future.

## Methods

**Plasmid constructs**. We generally used total cDNAs prepared from ALI d3 mTECs to PCR amplify cDNA fragments of centrosomal and ciliary proteins. The cDNAs encoding full-length mouse Centrin1 (NM_007593) and Pcm1 (NM_023662) were PCR-amplified and subcloned into pEGFP-N3 and pEGFP-C1, respectively. The RNAi-insensitive construct for GFP-Pcm1 was created by introducing silent mutations in siRNA-targeting regions. To generate pLV-APEX2, the GFP cDNA sequence in pLV-GFP was replaced by the APEX2 cDNA sequence amplified from pEGFP-APEX2-Tubulin (Addgene, 66171). The full-length cDNA for mouse Cep131 (NM_009734) was then PCR-amplified and subcloned into pLV-APEX2 to express APEX2-Cep131. To generate pLV-SNAP-Deup1, the fragment encoding GFP in pLV-*Deup1*-GFP-Deup1[26] was replaced by the fragment of SNAP amplified from pENTR4-SNAPf (Addgene, 29652).

To generate adenoviral expression constructs, APEX2-Cep131, Centrin1-GFP, GFP-Centrin1, and siRNA-insensitive GFP-Pcm1 were PCR-amplified and subcloned into the entry vector pYr-1.1 (YRBio, China). To generate adenoviral constructs to express shRNAs, DNA sequences encoding Ctrl-i and Pcm-i1 shRNAs were synthesized and inserted downstream of the U6 promoter of the entry vector pYr-1.1 expressing GFP-Centrin1. The LR recombination reactions between the entry constructs and the destination vector pAd/BLOCK-iT-DEST (YRBio, China) were performed using LR Clonase II enzyme mix (11791020, ThermoFisher).

To generate bacterial expression constructs for His-Pcm1 (1626-2025 aa, NM_023662), His-Cep135 (full-length, NM_199032), His-Cep120 (601-988 aa, NM_178686), His-Cep215 (1-400 aa, NM_145990), His-Odf2(401-826 aa, NM_001177659), His-Rootletin (1101-2009 aa, NM_172122), His-Pcnt (1485-2896 aa, NM_001282992), His-Cep131 (full length, NM_009734), GST-Cep162 (1-300 aa, NM_199316), GST-Cep290 (2001-2479 aa, NM_146009), and GST-Ofd1 (full length, NM_177429), the cDNA fragments encoding the indicated amino acids were amplified by PCR and subcloned into pET-32a/28a or pGEX-4T-1. The fusion proteins were expressed for antibody production or purification. Disordered regions of Pcm1 were predicted by using IUPred2A (https://iupred2a.elte.hu/plot)[60]. To express His-GFP-tagged Pcm1N (326-814 aa), Pcm1M (1072-1337 aa), and Pcm1C (1694-2024 aa) for phase separation assays, the cDNA fragments were PCR-amplified from the full-length cDNA and cloned into pET30a-GFP[61]. All constructs were verified via Sanger sequencing analysis. Primers used are listed in Supplementary Data 2.

**Antibodies**. His or GST tagged proteins were expressed using BL21 and purified with Ni-NTA agarose beads (Qiagen) or glutathione agarose beads (Sigma). To generate the Rat polyclonal Pcm1 antibody, immobilization with His-Pcm1 protein (1626-2025 aa) and subsequent affinity purification were carried out by GL Biochem (China). The rabbit polyclonal Cep131 antiserum was generated against His-Cep131 (full length) and affinity-purified (Immune Biotech, Shanghai). Rabbit polyclonal antibodies against His-Cep135 (full-length), His-Cep120 (601-988 aa), His-Cep215 (1-400 aa), His-Pcnt (1485-2896 aa), His-Rootletin (1101-2009 aa), GST-Cep290 (2001-2479 aa) and GST-Ofd1 (full length), and guinea pig antibodies against GST-Cep162 (1-300 aa), and His-Odf2(401-826 aa) were produced by ABclonal and affinity-purified.

All the antibodies used are listed in Supplementary Data 3.

**Cell culture and transfection**. HEK293T and HEK293A cells were grown in Dulbecco's modified Eagle's medium (DMEM) medium supplemented with 10% fetal bovine serum (FBS, Thermo Fisher) and 1% penicillin/streptomycin.

mTECs isolated from 4-week C57BL/6J mice and mEPCs from P0 mice were cultured as described previously[8,26,62]. After removing attached connective and muscle tissues, tracheas from 4-week C57BL/6J mice were chopped up along the vertical axis and digested in Ham's F-12K medium with 0.15% Pronase E (Sigma, P6911) and 0.1 mg/ml DNase I (Sigma, D5025) overnight at 4 °C. Released cells were collected by centrifugation for 5 min at $400 \times g$ at room temperature (r.t.) and resuspended with mTEC basic medium [DMEM-Ham's F-12 medium (Thermo Fisher, 11330-032) supplemented with 3.6 mM sodium bicarbonate, 4 mM L-glutamine, 1% penicillin/streptomycin, 0.25 µg/ml fungizone] with 10% FBS. After culturing at 37 °C for 4 h, the non-attached mTECs were collected by centrifugation at $400 \times g$ for 5 min, resuspended in mTEC plus medium [mTEC basic medium supplemented with 10 µg/ml insulin (Sigma, I6634), 5 µg/ml transferrin (Sigma, T8158), 0.1 µg/ml Cholera toxin (Sigma, C8052), 25 ng/ml epidermal growth factor (Sigma, E4127), 30 µg/ml bovine pituitary extract (Sigma, P1167), 5% FBS, and 0.05 µM retinoic acid (freshly added; Sigma, R2625)], and seeded into collagen (Sigma, C8897)-coated 6.5-mm Transwells with 0.4-µm-pore polyester membrane insert (Corning, 3470). When the cells reached full confluency, ALI was created by removing the medium in the Transwell insert and replacing the medium in the bottom compartment with the mTEC differentiation medium [mTEC basic medium supplemented with 2% Nu Serum (BD, 355100) and 0.05 µM retinoic acid (freshly added)] to induce differentiation. DAPT (Sigma, D5942) was added to 10 µM at day 1 post ALI to increase the MCC differentiation efficiency.

For mEPCs, P0 mice telencephala were chopped up after removing the cerebellum, olfactory bulbs and meninges with sharp tweezers (Dumont, 1214Y84) in cold dissection solution (161 mM NaCl, 5 mM KCl, 1 mM $MgSO_4$, 3.7 mM $CaCl_2$, 5 mM Hepes, and 5.5 mM Glucose, pH 7.4) under a stereo microscope. The telencephala were digested with 1 ml of the dissection solution containing 10 U/ml papain (Worthington, LS003126), 0.2 mg/ml L-cysteine, 0.5 mM EDTA, 1 mM $CaCl_2$, 1.5 mM NaOH, and 0.15% DNase I (Sigma, D5025) for 30 min at 37 °C. Released cells were mechanically dissociated by pipetting up and down 10 times with a 5-ml pipette and collected by centrifugation at $400 \times g$ for 5 min at r.t. Cells were resuspended with DMEM medium supplemented with 10% fetal bovine serum (FBS) and 1% penicillin/streptomycin, and inoculated into the laminin-coated flask (Sigma, L2020). Differentiated oligodendrocytes and neurons were mechanically removed by striking the flask roughly 2 days after inoculation. Cells were trypsinized and inoculated into laminin-coated glass-bottom dishes when cells reach confluence. After cells cultured on glass-bottom dishes had grown to full confluence, FBS was removed from the medium to initiate differentiation.

For transfection of plasmids, cells were transfected at ~70% confluency with Lipofectamine 2000 (Thermo Fisher). For transfection of siRNA oligos, mEPCs were transfected with the siRNA oligos and Lipofectamine RNAiMAX transfection reagent (Thermo Fisher). In order to enhance the knockdown efficiency, mEPCs were continuously transfected with each siRNA oligo every 3 days. siRNA oligos were purchased from GenePharma (China). The sequences are: NC (Ctrl i): 5′- TT CTCCGAACGTGTCACGTtt-3′; Pcm1i1: 5′-GCACCAGGAAUGAAUUUCAtt-3′; Pcm1i2: 5′-GACCCAACAACAGUAACUAtt-3′.

Experiments involving mouse tissues were performed in accordance with protocols approved by the Institutional Animal Care and Use Committee of Institute of Biochemistry and Cell Biology.

**Viral production and infection**. Lentiviral particles were produced and used to infect mEPCs as described previously[26]. HEK293T cells were transfected with the lentiviral plasmids, packaging plasmid (Delta 8.9) and envelope plasmid (VSV-G) at a ratio of 5:3:2 for 48 h. Supernatant containing the lentiviral particles was used to infect ependymal progenitors 2 days ahead of serum starvation. Adenovirus packaging was carried out according to the manufacturer's instructions (YRBio, China). In brief, the adenoviral expression constructs were digested with *Pac*I restriction enzyme (NEB). The linearized adenoviral expression construct was transfected into HEK293A cells in 6-well plates at 70% confluency. Culture medium was collected as the adenoviral stock when most cells became round and detached from dishes. Adenovirus was amplified by infecting fresh HEK293A cells with the adenoviral stock. After approximately 80% cells showed a cytopathic effect, cells were harvested and virus were released by three freeze-thaw (−80 and 37 °C) cycles. The harvested adenoviral particles were used at a 1:1000 or 1:200 dilution.

To transduce the adenoviral constructs into mEPCs, the adenovirus-containing medium was added into the culture medium of mEPCs. To infect mTECs, the cells cultured in the Transwell insert were rinsed twice with PBS and incubated with 12 mM EGTA in 10 mM Hepes (pH 7.4) for 20 min at 37 °C to disrupt the tight junction, followed by washing once with PBS. The insert was supplemented with the adenovirus-containing medium and centrifuged at $400 \times g$ for 80 min at 32 °C. The cells were cultured for 24 h and then supplemented with fresh culture medium.

**Immunofluorescent microscopy**. Immunofluorescence microscopy was carried out as described[26]. mEPCs grown on glass-bottomed dishes (Cellvis, D29-14-1.5-N) and mTECs in Transwells (Corning, 3470) were pre-permeabilized with 0.5% Triton X-100 in PBS for 40 s and 3 min, respectively, to remove soluble proteins, followed by fixation with 4% fresh paraformaldehyde in PBS for 15 min at r.t. After fixation, cells were extracted with 0.5% Triton X-100 in PBS for 15 min and blocked with blocking buffer (4% BSA in TBST) for 1 h at r.t. Primary and secondary antibodies or streptavidin conjugated with Alexa Fluor 568 were prepared in blocking buffer and applied to cells at r.t. for 2 h and 1 h, respectively. Anti-GFP antibody and Alexa Fluor-488-conjugated secondary antibody were used to enhance GFP fluorescent signals in fixed cells. Super resolution images were

obtained with a 3D structured illumination microscope at 0.125 μm intervals (GE DeltaVision OMX Imaging system). Raw images were processed for maximum intensity projection with SoftWoRx software. The confocal images were acquired using a Leica TCS SP8 confocal platform with an HCX PL APO 63×/1.4 oil immersion objective (Deerfield, IL) with a four-line mean averaging protocol. Optical sections were captured at 0.5 μm intervals and z-stack images were obtained with maximum intensity projections (Leica Microsystems, Germany).

**Live cell imaging**. Live imaging of ependymal ciliary motions was carried out at 7-ms intervals by using a high-speed digital camera (Andor Neo sCMOS) on an inverted microscope (Olympus IX71) with a 63×/1.40 oil immersion objective. The ciliary trajectories were established by manually tracking the ciliary tip positions frame by frame with ImageJ (Fiji). Four recognizable cilia of each cell were analyzed over a period of 56 ms[24]. Kymographs were generated for the first 1.4 s using ImageJ.

As mTECs were grown on opaque polyester membranes, bright-field microscopy was not suitable for imaging their multicilia motility. mTECs grown in Transwells were thus incubated with 200 nM SiR-tubulin (Spirochrome, YS-SC002) for 1 h to fluorescently label their multicilia. The polyester membrane was cut around the membrane edges with a scalpel. The membrane was put upside down in a glass-bottomed dish in medium. The cells on the membrane were imaged with a spinning disk confocal super-resolution microscope (Olympus SpinSR10) equipped with an APON 60× OTIRF/1.49 NA oil objective (Olympus) and an ORCA-Flash 4.0 V3 digital CMOS camera (Hamamatsu). To locate adenovirus-infected cells, the autofluorescence of GFP-Centrin1 was firstly captured at the plane of basal bodies. The focal plane was then shifted to the multicilia, where a single-layer autofluorescence of SiR-tubulin was captured at 15-ms intervals for 3 s. The laser power for the SiR-tubulin channel (640 nm) was set to 50% to achieve an exposure time of 14 ms. Kymographs were generated for the first 3 s using ImageJ.

To image FGM dynamics, mouse ependymal progenitors were infected with adenovirus at day −2 to express GFP-Pcm1 and lentivirus at day −1 to co-express SNAP-Deup1 under *Deup1* promoter[26]. Live cell imaging was performed at day 3 with the spinning disk microscope after incubating with 200 nM SNAP-Cell TMR-Star for 30 min. Laser powers were set to 15% (488 nm) and 10% (561 nm) to reduce cell toxicity. Images were recorded at 5-min intervals for 65 min with an exposure time of 100 ms for either channel. z-stack sectioning was performed at 0.5-μm intervals to cover a depth of 8.5 μm. The images and movies were processed with Imaris (Bitplane) and ImageJ.

For FRAP assays, mouse ependymal progenitors were infected with adenovirus at day −1 to express GFP-Pcm1. FRAP assays were performed at day 3 following the FRAP wizard in a Leica SP8 confocal microscope system. GFP-Pcm1 puncta were photobleached using the 488-nm laser at 50% laser power for 1.3 s twice, followed by imaging at 20 s intervals at 1% laser power for 10 min.

**Electron microscopy**. To visualize FGMs, transmission EM and immune-gold labeling were carried out as described[8]. mTECs grown on transwells at day 3 postALI or mEPCs on glass-bottomed dishes at day 3 postserum starvation were pre-permeabilized with 0.5% Triton X-100 in PBS for 3 min or 40 s, respectively. The cells for conventional transmission EM or FIB-SEM were fixed in buffer A (150 mM cacodylate buffer containing 2.5% glutaraldehyde, 4% paraformaldehyde, and 2 mM CaCl₂) for 15 min at r.t. For immuno-EM, the pre-extracted mTECs were fixed in 4% fresh paraformaldehyde in PBS for 15 min at r.t., followed by extraction with 0.5% Triton X-100 in PBS for 15 min and incubation with blocking buffer (4% BSA in TBST) for 1 h. The cells were incubated with primary antibodies in the blocking buffer overnight at 4 °C and for an additional 4 h at r.t. After incubation with nanogold-conjugated secondary antibodies for 1 h at r.t., the cells were further fixed in buffer A for 15 min at r.t. The fixed samples were incubated with 150 mM cacodylate buffer containing 1.5% K₄Fe(CN)₆, 2% OsO₄, and 2 mM CaCl₂ for 30 min on ice, 1% thiocarbohydrazide for 20 min at r.t., 2% OsO₄ for 30 min at r.t., 1% uranyl acetate overnight at 4 °C, and 1% Pb(NO₃)₁ in 30 mM aspartic acid (pH 5.5) for 30 min at 60 °C. The samples were then dehydrated through an ethanol series, gradually infiltrated through an ascending series of low viscosity ethanol-resin mixture (25, 50, 75, and 100% Epon 812 resin) and polymerized at 60 °C for 48 h. For mEPCs, hydrofluoric acid was applied to dissolve the glass of the culture dish after polymerization. The resin block was air-dried and pushed away from the dish. Ultrathin sections of approximately 70-nm thickness were sectioned from the resin blocks using a Leica Ultracut UCT ultramicrotome and collected on metal mesh grids for transmission EM.

The resin blocks for FIB-SEM were trimmed to obtain a smooth flat surface where regions of interest were exposed, and the samples were glued on a stage and sputtered with gold (Hitachi E1010) for 90 s. Serial FIB milling and SEM imaging were performed by consecutively milling the block surface with gallium ion beam followed by imaging using an electron beam with 2 kV acceleration voltages, 0.4 nA current, and 10 μs dwell time on a FEI Helios NanoLab G3 UC FIB-SEM. The serial z-stack sectioning was carried out at 5-nm intervals with a 10.1-μm horizontal field of view and imaged at a magnification of 27,246× with a resolution of 4096 × 3536 pixels. The SEM images were aligned and processed for 3D reconstructions using Imaris (Bitplane). The structural segmentation and 3D surface generation were performed using the segmentation module in Amira 6.0 (Thermo Fisher). The movies of structural models were generated using Fiji[63].

For transmission EM of the ciliary ultrastructure, cultured mEPCs at day 10 were fixed in PBS containing 2.5% glutaraldehyde and 4% paraformaldehyde for 30 min at 37 °C without pre-extraction, followed by treatment with 0.15% tannic acid in PBS for 1 min and 2% OsO₄ for 1 h at 4 °C. The samples were then dehydrated, infiltrated with Epon 812 resin, and polymerized. The 70-nm ultrathin sections were poststained with 2% uranyl acetate for 10 min and 1% lead citrate for 5 min, followed by imaging at 80 kV using a Tecnai G2 Spirit transmission electron microscope (FEI).

**Proximity labeling**. Biotin-phenol labeling was performed as described[29] with minor modifications. mEPCs expressing APEX2-Cep131 were incubated with fresh medium containing 500 μM biotin-phenol (Iris Biotech GmbH) at 37 °C for 30 min. One millimolar of H₂O₂ in fresh medium was applied with gentle agitation to initiate the 1 min biotinylation reaction. Cells were then washed with PBS containing 10 mM sodium ascorbate, 10 mM sodium azide, and 5 mM Trolox three times to quench the reaction, and subsequently subjected to immunostaining.

To enrich the biotinylated proteins, the mEPCs treated with biotin-phenol were lysed in RIPA buffer supplemented with 1 mM PMSF, 1 mM dithiothreitol (DTT), and the proteinase cocktail (Sigma, 539134). The lysates were sonicated and cleared by centrifugation at 15,000 × g for 10 min at 4 °C. The supernatants were incubated with Streptavidin agarose beads for 4 h at 4 °C. The beads were washed with RIPA buffer three times. The bound proteins were eluted with 2× protein loading buffer supplemented with 20 mM DTT and 2 mM biotin and boiled for 5 min. The eluted samples were subjected to mass spectrometry analysis.

**Liquid droplet formation and hydrogel formation**. His-GFP-tagged Pcm1N, Pcm1M, and Pcm1C were expressed in *E. coli* BL21(DE3) strain for 20 h at 16 °C, in the presence of 1 mM IPTG. The bacteria were lysed in ice-cold lysis buffer (50 mM NaH₂PO₄, 500 mM NaCl, 10 mM imidazole, 0.5% Triton, 10% glycerol, and pH 8.0) containing 1 mM PMSF, 1 mM DTT and protease cocktail (Sigma, 539134) by using a high-pressure homogenizer (JN-02C, JNBIO). The proteins were absorbed on Ni-NTA beads, eluted using 300 mM imidazole following the manufacturer's handbook (Qiagen), and dialyzed into the P buffer (20 mM Tris-HCl, pH 8.0, 150 mM NaCl, 10% glycerol, and 1 mM DTT) at 4 °C overnight. They were further concentrated to 500 μM for His-GFP-Pcm1N and His-GFP-Pcm1C and 300 μM for His-GFP-Pcm1M using Amicon Ultra 30 K centrifugal filter devices (Millipore). 5-μl or 10-μl protein aliquots were snap-frozen in liquid nitrogen and stored at −80 °C.

Liquid droplet formation was performed as described[51,52]. Five microliter of the proteins at varying concentrations, with or without 5% polyethylene glycol 3350 (PEG3350), was incubated at 25 °C for 2 min. After the incubation, 3 μl of each mixture were loaded into a small chamber generated by sticking a glass coverslip to a glass slide using a pair of double-sided tapes and imaged under a widefield fluorescent microscope (Olympus IX71) with a 60× oil immersion objective. FRAP assays were performed following the FRAP wizard installed in a Leica SP8 confocal microscope system. Liquid droplets of His-GFP-Pcm1M were photobleached using a 488-nm laser with 50% power for four times, 1.3 s each time, followed by imaging at 2-s intervals with 1% laser power for up to 2 min.

Hydrogel formation was performed as described[53]. 50 μl of 100 μM His-GFP-Pcm1N or His-GFP-Pcm1C in a 1.5 ml tube were topped with 100 μl freshly prepared P buffer. After centrifugation at 15,870×g for 30 min at 4 °C, the tubes were photographed directly against a white striped background to show the transparency of the pellets. After removing the supernatants from the tubes, the pellets were incubated at 4 °C for an additional 24 h to solidify. The solidified transparent hydrogels were carefully removed from the tubes and placed on glass coverslips for photography.

**Quantification and statistical analysis**. The numbers of centrioles and deuterosomes, and the deuterosome diameters were measured from 3D-SIM images using default setting of the "count/size" function and "automatic bright objects" mode of Image-Pro Plus 6.0 software (Media Cybernetics) as described previously[26]. The percentage of MCCs were calculated by counting the numbers of the DAPI-positive nuclei and acetylated tubulin positive cells from confocal images. To quantify motility patterns, MCCs with their cilia beating predominantly in a back-and-forth manner were scored as "planar", whereas those predominantly containing immotile and/or irregularly-beating cilia were scored as "abnormal". To measure ciliary beat frequencies, the total time of ten beat cycles of traceable cilia was manually measured, followed by conversion of the results to cycles per sec (Hz). Quantification results are presented as mean ± s.d. Samples that passed the normality test were subjected to a two-tailed unpaired Student's *t*-test using Graphpad Prism software, otherwise the non-parametric two-tailed Mann–Whitney *U*-test was used to determine statistical significance. Differences were considered significant when *P* was <0.05.

**Reporting summary**. Further information on research design is available in the Nature Research Reporting Summary linked to this article.

## Data availability
Data that support the results of this study are available from the corresponding author upon reasonable request. Source data are provided with this paper.

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

## Acknowledgements

The authors thank Xiaomin Li (Tsinghua University Cryo-EM Facility), Wenjuan Cai (Olympus Corporation), Qi Gao, the Center for Biological Imaging (Institute of Biophysics, CAS), mass spectrometry and integrated laser microscopy systems (National Facility for Protein Science Shanghai), and institutional core facilities for cell biology and molecular biology for instrumental and technical supports. This work was supported by National Key R&D Program of China (2017YFA0503500), National Natural Science Foundation of China (31991192, 31771495, and 31501092), and Chinese Academy of Sciences (XDB19020102).

## Author contributions

X.Z. and X.Y. conceived and directed the project; H.Z. and Q.C. performed major experiments; F.L. performed in-vitro phase separation assays; L.C. and X.L. performed structural segmentation and modeling for the FIM-SEM data; L.X. contributed to quantifications; Q.H. contributed to homemade antibodies; J.Z. provided the 3D-SIM system; X.Z., X.Y., H.Z. and Q.C. designed experiments, interpreted data, and wrote the paper.

## Competing interests

The authors declare no competing interests.
