## [Peer Review File · Nature Communications]

REVIEWER COMMENTS

Reviewer #1 (Remarks to the Author):

In their recent submission, Zhao et al. demonstrate that fibrogranular materials (FGMs), largely organized by Pcm1, selectively concentrate several centriole-related proteins required for proper multiciliated cell ultrastructure in a cytosolic phase-separated compartment. Using cell culture, proximity labelling and mass spec, and a series of beautiful imaging experiments, the authors convincingly describe very particular set of centriolar proteins localized within FGMs during the differentiation of MCCs. Using siRNA they show that most of this organization is lost upon depletion of Pcm1, however ciliation is not markedly reduced. Instead depletion of Pcm1 lead to a marked reduction in ciliary motility, altered deuterosome distribution, size, and number, and severely disrupted ciliary ultrastructure specifically effecting the central pair. Finally, the authors touch on the FGMs being a phase-separated compartment by observation of fusion of particles in cell culture and in vitro, and showing that they can form hydrogels upon centrifugation. While the authors overall provide a thorough and convincing manuscript, there are several concerns to address prior to publication in Nature Communications. These concerns center around a lack of quantitation that is apparent throughout the manuscript that can be easily remedied and would greatly bolster the arguments made in these findings.

Concerns:

1. The authors state in figure 1 the size and number of Pcm1+ condensates changes along differentiation but provide no quantitation of this. A simple size vs. number graph would show this nicely along the differentiation timeline.
2. The authors use siRNA regularly throughout the manuscript, while they do provide a control by reintroducing a rescue construct to show specificity, I have some concerns with differences observed between siRNA and actual genetic mutant analysis. Could authors provide an orthologous method (perhaps CRISPR in their cell culture model) and reproduce similar results?
3. While the authors provide beautiful images to show GFP-Pcm1 rescues FGMs, they do not show any quantitation to show that rescue is significant. They quantify deuterosome size and number in their Pcm1 siRNA experiments, a similar quantitation in the rescue experiments would be appreciated.
4. While the authors do an excellent job showing the ultrastructure defects in the axonemes of cilia post Pcm1 depletion in figure 3, the images in Figure3d are wildly uninformative. Perhaps showing kymographs for ciliary beat would be more convincing, or still frames of subsequent frames for each motion. A different way to represent 3d would be greatly appreciated.
5. Additionally, in figure 3 the authors simply quantify % multiciliated cells, not % cilia/cells or basal bodies/cell. Is there a reduction in cilia/ cell or basal bodies/cell in case of loss of Pcm1?
6. The authors attempt to show the proteins identified in their PL-MS samples colocalize with Pcm1 in FGMs, but only show single frames in gray scale of Pcm1 and/or each hit in 4d, we suggest showing merges of these would be much more convincing to show these hits localize within FGMs with Pcm1.
7. The authors claim loss of Pcm1 causes excess of premature centriolar targeting of proteins but only show images in Figure 6, could this be quantified?
8. Finally, while the authors provide some beautiful images of fusion of the GFP-Pcm1 positive foci, one hallmark of phase separation, but they do not show how quickly these particles can turn over within the particles and from the cytosol. How quickly is material exchanged from within the particle? And from the cytosol?

Minor comments:

1. Figure 1e looks as though a black smear has been placed over the top left corner to add in an XY axis, this should be removed and the axis placed accordingly on image so image is not altered in such a way
2. Figure 5 is loaded with images, the small insets on sides of main images are in gray scale and with no labelling of which protein each gray scale image is for, suggestion for colors to be kept similar between main images and small cropped images or a clarification in legend or in figure of what each

gray scale image is showing.

Reviewer #2 (Remarks to the Author):

Review of Zhao et al., 2020

The authors provide a detailed description of the role of fibrogranular material in centriole assembly and motile cilia function in multiciliated cells. Fibrogranular material has been observed in multiciliated cells for many years by electron microscopy, but its function has not yet been explored. This manuscript provides strong evidence that Pcm1 is an essential component of fibrogranular material, and that the absence of fibrogranular material results in deuterosomes that are more numerous, but smaller in size. Additionally, motile cilia assembled in cells without fibrogranular material have structural and functional defects. Overall, I felt this manuscript provided an excellent characterization of fibrogranular material and introduces novel components of this structure that will be of significant interest to the centriole and motile cilia fields. The manuscript is clearly presented, and the quality of the data is generally very high. Below, I make a number of recommendations that should help improve this already excellent work. Many of these changes can be made by modifying the text or with additional analysis of the available data.

Major points:

1. On lines 104-105, the authors write, "Interestingly, deuterosomes still formed massively and produced ¹⁰⁰procentrioles in the Pcm1-depleted cells (Fig. 2c). " However, there was no quantification of centriole or cilia number in the Pcm1-depleted cells. This reviewer believes it is essential to know if Pcm1-depleted cells amplify similar numbers of centrioles to control cells.
2. Related to point 1, the authors' claim that deuterosomes still form following Pcm1 depletion is based on Cep152 immunofluorescence staining. However, it would be very helpful to stain deuterosomes directly using an antibody against Deup1. This will allow direct characterization of deuterosomes in control and Pcm1-depleted cells.
3. In figure 2g-h, the authors quantified deuterosome number and size and found that in Pcm1-depleted cells, deuterosome number increased while size decreased. It would be useful to also quantify if the number of procentrioles associated with each deuterosome decreases in Pcm1-depleted cells.
4. The labels in figure 2 are confusing. The quantifications in Figures 2g and 2h correspond to the siRNA experiments in figure 2d. Therefore, they should be relabeled as 2e and 2f, respectively. The rescue experiments shown in figure 2e and 2f should come last in the figure.
5. The rescue experiment in figure 2e,f would benefit from more characterization. I appreciate that TEM analysis on the Pcm1-rescue cells would be challenging due to the low transduction efficiency of the adenovirus expressing the rescue construct. However, the authors could test if the rescue construct is able to rescue deuterosome size and number. Centriole number in MCCs with rescued Pcm1 should also be reported.
6. It is important to provide more information on motile cilia function in Pcm1-depleted cells. In figure 3d, the authors quantify planar versus abnormal cilia beating patterns. In the materials and methods,

the authors should clearly explain what qualifies as an “abnormal” beating pattern. If just one cilium moves in a circular motion, does that qualify the cell as “abnormal,” or must all cilia be beating in a circular motion? In addition, cilia beat frequency should be measured in the control and Pcm1-depleted cells as an additional determination of cilia function.

7. In Figure 7, the authors describe the condensate properties of fibrogranular material. However, in 7b, the GFP-Pcm1 droplets look completely different from the Pcm1 structures shown throughout the rest of the manuscript (for example, in figure 1, 2c, and 2f). This reviewer believes that the droplet-like localization of GFP-Pcm1 shown in figure 7b could be an artifact of overexpression: something that is common with overexpressed centrosome/centriole proteins. In the absence of monitoring Pcm1 behavior with an endogenous tag, I would strongly recommend removing figure 7b from the manuscript as this data could be very misleading. Instead, the authors should focus on the in vitro phase separation data supported by the disordered regions of the protein.

Minor points

1. Detailed methods for ependymal cell cultures and mTEC cultures should be described.

2. In figure 1b-c, the authors show that throughout stages of differentiation, the number of Pcm1 condensates decreases, but their size increases. If possible, this could benefit from quantification.

3. The representative TEM images in figure 2d for control and Pcm1-depleted cells look very different. The deuterosomes are much darker in the Pcm-i1 and the procentrioles are not easy to distinguish. Can the authors comment on this apparent difference? Also, a representative 1 μm image should be shown for the control sample to match the Pcm-i1 sample.

4. Relating to figure 3, in lines 148-150 in the text the authors state “In addition, 53% of ciliary transition zones ($n = 109$) and 52% of the basal bodies ($n = 68$) in the Pcm-i1-treated cells also displayed defects in microtubule organization, comparing to the 100% normal ultrastructure ($n \geq 26$) in the control cells (Fig. 3k,l).” The authors should show the quantifications for the transition zone and basal body sections of the cilia similar to how they showed these quantifications for the axoneme regions in figure 3j.

Reviewer #3 (Remarks to the Author):

This study by Zhao and colleagues investigates the role of electron dense fibrogranular material (FGM) in multiciliated cells. They demonstrate that the centriolar satellite protein PCM1 is a core component of FGMs, and helps to form these structures via phase separation. The authors show that FGMs are distinct from deuterosomes and centrioles, but can localize and concentrate a number of centriole and cilia-associated proteins. Proteomic analysis of FGM proteins identified a number of centriolar and ciliary proteins, in addition to other cellular components. Disruption of FGMs via depletion of PCM1 in multiciliated cells caused abnormal centriolar targeting of FGM client proteins, which then resulted in altered deuterosome size, number, and distribution. Even though cilia formation was not impacted, the cilia showed structural defects that disrupted their motility. In sum, the authors conclude that FGMs in multiciliated cells help to concentrate deuterosome and centriole-related proteins, necessary for proper assembly of basal bodies and motile ciliary axonemes.

Overall, the story is very intriguing and somewhat consistent with prior data. However, I believe the manuscript suffers from a number of experimental flaws that makes it difficult for this reviewer to accept the interpretations of the results presented in this study. As such, I do not believe it is ready for publication in its current form.

Major concerns:

- One of the main issues is that the authors switch between airway multiciliated cells and ependymal multiciliated cells throughout the paper, performing different assays in each (detailed more below). Although this is presented as a strength, it is in fact a weakness because the biology of these two cell types is not the same, even though both are multiciliated cells. In recent years it is becoming increasingly evident that the kinetics of centriole amplification, centriole and cilia number, and organization of these structures is different between these two cell types. This is also evident when considering human disease phenotypes: there are many mutations that impact ciliary motility where patients present with respiratory defects (e.g. primary ciliary dyskinesia) due to defects in airway multiciliated cells, but without any ependymal cell dysfunction. Similarly, there are mutations that impact ependymal ciliary motility (causing hydrocephalus) that do not cause respiratory defects, and those patients have normal motile cilia function in airway multiciliated cells. Thus, at the cellular and organ level there are subtle differences in the biology of these cells, therefore correlating findings from one cell type to the other is not accurate in many cases.

- This issue is highlighted in Fig 1 and Supp Fig 1: in Fig 1 the authors do a convincing job of demonstrating that PCM1-containing FGMs are DISTINCT from deuterosomes and centrioles in airway multiciliated cells - using a combination of SIM and electron microscopy, they show that these FGMs form circular-looking structures by SIM that are adjacent to, but not localized to, either deuterosomes and centrioles (this point is emphasized throughout the paper). This is also confirmed by immune-gold labelling and TEM analysis of FGMs. However, when the authors try to confirm this finding in ependymal MCC, it is very obvious that PCM1-containing structures are localized differently in these cells. At almost every stage of centriole amplification and differentiation, the distribution of PCM1 and its relative position to deuterosomes and centrioles is different in the two cell types. There is a lot of PCM1 associated with both deuterosomes and procentrioles, and they are distributed in more puncta than the strictly circular looking FGMs in airway MCC. The authors use arrows to show a few examples of PCM1 not localizing to deuterosomes or procentrioles, yet the majority of PCM1 is in fact associated with those structures - I can clearly see many PCM1 positive deuterosomes and procentrioles, as well as the nuclear envelope enrichment (again not seen in airway MCC). Hence, the conclusion that PCM1 is in FGMs and generally is distinct from deuterosomes and newly forming centrioles is inaccurate. This is all important because the functional studies using depletion of PCM1 are all subsequently performed in ependymal MCC, where this is clearly not true.

- Next, the authors deplete PCM1 to study FGM function, but they do this only in ependymal MCC using RNAi. They note a change in deuterosome number and size, but not centriole number or ciliogenesis (as was shown previously). Importantly, they note a defect in basal body morphology (which is not quantified) and ciliary ultrastructure. It makes sense to me that loss of PCM1 might cause centriolar defects in ependymal MCC, since the protein is associated with the newly forming basal bodies. But what about in airway MCC where it is distinct from these structures, a major point of Fig 1? For this reviewer to be convinced regarding the observed defects upon PCM1 depletion, the authors will need to inhibit the formation of FGMs in airway MCC and show the same phenomenon (basal body and cilia ultrastructural defects, abnormal motility) occurs there. Otherwise, the interpretation of this (and subsequent) experiment is inaccurate.

- The authors then perform proximity labeling proteomics using a PCM1 interactor, Cep131, and do this in ependymal MCC. They identify a host of proteins, some of which are associated with centrioles and cilia. To make the case that these centriolar proteins are associated with FGMs, they go back to airway MCC and do localization studies. Here again the same issue presents itself: for example, they suggest that Cep135 (a core cartwheel associated protein essential for procentriole nucleation) and Cep120 (a core centriolar protein necessary for their growth and elongation) are both found enriched in FGMs and are distinct from deuterosomes and procentrioles. Yet this is contradictory to the many published papers (including previously from this lab) showing these proteins are in fact associated with procentrioles during centriole amplification in airway MCC (also noted in Fig. 5). How are

proteins, such as Cep135 and Cep120, that are normally enriched on procentrioles during their formation, and which has been shown by multiple groups, now be excluded from procentrioles during centriole amplification?

- Fig 5 – since the identification of these centriolar “client” proteins was done in ependymal MCC using proximity labeling with Cep131 as bait, why did the authors move to airway MCC to validate their localization to FGMs?! I suspect it’s because of the same issues described above.
- Although I am not as familiar with phase separation dynamics, most of the studies I have seen show much higher kinetics in rates of fusion/fission events. The live-cell imaging studies (Fig 7b) show these events happening on the scale of 20-25 minutes. Is this normal for phase separated structures?

Responses to the comments of reviewers

Reviewer #1:

In their recent submission, Zhao et al. demonstrate that fibrogranular materials (FGMs), largely organized by Pcm1, selectively concentrate several centriole-related proteins required for proper multiciliated cell ultrastructure in a cytosolic phase-separated compartment. Using cell culture, proximity labelling and mass spec, and a series of beautiful imaging experiments, the authors convincingly describe very particular set of centriolar proteins localized within FGMs during the differentiation of MCCs. Using SiRNA they show that most of this organization is lost upon depletion of Pcm1, however ciliation is not markedly reduced. Instead depletion of Pcm1 lead to a marked reduction in ciliary motility, altered deuterosome distribution, size, and number, and severely disrupted ciliary ultrastructure specifically effecting the central pair. Finally, the authors touch on the FGMs being a phase-separated compartment by observation of fusion of particles in cell culture and in vitro, and showing that they can form hydrogels upon centrifugation. While the authors overall provide a thorough and convincing manuscript, there are several concerns to address prior to publication in Nature Communications. These concerns center around a lack of quantitation that is apparent throughout the manuscript that can be easily remedied and would greatly bolster the arguments made in these findings.

Response:

We thank our reviewer for appreciating our efforts. We have included the requested quantification results in the revised manuscript and addressed other concerns. We hope that the revised manuscript will be satisfactory to our reviewer. We would also like to thank our reviewer for helping us to improve the manuscript.

Concerns:

1. The authors state in figure 1 the size and number of Pcm1+ condensates changes along differentiation but provide no quantitation of this. A simple size vs. number graph would show this nicely along the differentiation timeline.

Response:

The requested quantification results have been included in the revised manuscript (Fig. 1d and 1e).

2. The authors use siRNA regularly throughout the manuscript, while they do provide a control by reintroducing a rescue construct to show specificity, I have some concerns with differences observed between siRNA and actual genetic mutant analysis. Could

authors provide an orthologous method (perhaps CRISPR in their cell culture model) and reproduce similar results?

Response:

Following the request, we explored the possibility to knockout *Pcm1* in mEPC precursor cells derived from P0 *mCherry-Cas9* transgenic mice by transfecting *in-vitro* transcribed sgRNAs. Nevertheless, we were unable to observe convincing *Pcm1* depletion in the cells by immunostaining and immunoblotting and have attributed this to low knockout efficiency of the poorly proliferative mEPC precursors.

Fortunately, we notice that a poster in the 2019 Cold Spring Harbor Asia Conference on Cilia and Centrosomes presented studies on *Pcm1* knockout mice (please refer to the abstract in **Fig. 1 for reviewers**). The authors reported that *Pcm1* knockout mice are viable but develop severe ciliopathy phenotypes including hydrocephaly and male infertility, which are typical phenotypes of primary ciliary dyskinesia (PCD). Their results obviously strongly support ours and would sufficiently address the concerns of our reviewer.

Furthermore, we depleted *Pcm1* in mTECs and have presented the results in the revised manuscript. We observed similar influences on deuterosomes and ciliary motility in mTECs (Supplementary Fig. 3a-3c, 3f, 3g and Movie 4) as in mEPCs. Similar to the *Pcm1* depletion in mEPCs, the *Pcm1* depletion in mTECs also did not markedly affect the basal body numbers (Supplementary Fig. 3d and 3e). The depletion of *Pcm1* also caused premature centriolar targeting of Cep135 and elongated Cep131 streak in mTECs (Supplementary Fig. 6).

Based on these, we hope that our reviewer would agree that knocking out *Pcm1* in mEPCs is not an essential experiment for our current manuscript.

We'll keep exploring the possibility of establishing an efficient gene knockout protocol for cultured multiciliated cells in the future. mEPC precursors can only be cultured *in vitro* for a few days before they lose the potency to differentiate into multiciliated cells (please refer to Fig. 2a for an experimental scheme) (Spassky et al., 2005; Shah et al., 2018). They also need to grow into confluency to form a layer of epithelial cells with tight junctions before they can be induced to differentiate. Even if the rare knockout cells could be selected through drug selection, they were

Figure 1 for reviewers

unlikely able to expand into populations sufficient for the multiciliated cell differentiation. Therefore, we think we need to test different gene editing tools to find one that can efficiently work in these poor proliferating cells.

3. While the authors provide beautiful images to show GFP-Pcm1 rescues FGMs, they do not show any quantitation to show that rescue is significant. They quantify deuterosome size and number in their Pcm1 siRNA experiments, a similar quantitation in the rescue experiments would be appreciated.

Response:

We thank our reviewer for the comments. We have included the requested quantification results in the revised manuscript (Fig. 2j and 2k).

4. While the authors do an excellent job showing the ultrastructure defects in the axonemes of cilia post Pcm1 depletion in figure 3, the images in Figure3d are wildly uninformative. Perhaps showing kymographs for ciliary beat would be more convincing, or still frames of subsequent frames for each motion. A different way to represent 3d would be greatly appreciated.

Response:

We thank our reviewer for the comments. We have included kymographs at two representative positions in each image in the revised manuscript (Fig. 3e). We also quantified ciliary beat frequencies and have presented the results in Fig. 3g.

5. Additionally, in figure 3 the authors simply quantify % multiciliated cells, not % cilia/cells or basal bodies/cell. Is there a reduction in cilia/ cell or basal bodies/cell in case of loss of Pcm1?

Response:

As it is not practical to clearly count the number of multicilia in images of light microscopes, we chose to only count the number of basal bodies per cell and observed that the depletion of Pcm1 did not markedly affect the basal body production in mEPCs. The results have been presented in Fig. 3d in the revised manuscript. Similar results were obtained in mTECs when we depleted Pcm1 through an adenovirus-mediated RNAi (Supplementary Fig. 3d and 3e), consistent with the previous report (Vladar et al., 2007).

6. The authors attempt to show the proteins identified in their PL-MS samples colocalize with Pcm1 in FGMs, but only show single frames in gray scale of Pcm1 and/or each hit in 4d, we suggest showing merges of these would be much more convincing to show these hits localize within FGMs with Pcm1.

Response:

In the revised manuscript, we have included the merged images in Fig. 4d.

7. The authors claim loss of Pcm1 causes excess of premature centriolar targeting of proteins but only show images in Figure 6, could this be quantified?

Response:

In the revised manuscript, we have included the quantification results for procentriolar intensities of Cep135, Cep120, and Ofd1 in Fig. 6d. Consistently, the procentriolar intensities of Cep135, Cep120, and Ofd1 were weak in stage II and increased in later stages in the control mEPCs but became strong in all the Pcm1-depleted cells in stages II-IV (Fig. 6d). Although the Pcm1-depleted cells in stages II and III were indistinguishable, the mixed populations lacked procentrioles with weak centriolar Cep135, Cep120, or Ofd1 that resembled those in the stage-II control cells, suggesting a deregulated procentriolar targeting in stage II.

For better readability and clarity, we have included a summary on FGM localizations in Fig. 6c and moved the two sets of 3D-SIM images for Cep120 and Ofd1 to Supplementary Fig. 5a and 5b, respectively. Images of a representative stage-IV control cell are added to the Ofd1 panel (Supplementary Fig. 5b) to make the panel match with others.

8. Finally, while the authors provide some beautiful images of fusion of the GFP-Pcm1 positive foci, one hallmark of phase separation, but they do not show how quickly these particles can turn over within the particles and from the cytosol. How quickly is material exchanged from within the particle? And from the cytosol?

Response:

We performed the requested FRAP assays on the GFP-Pcm1 positive foci and have presented the results in Fig. 7c and 7d in the revised manuscript. We observed that the FGM condensates are dynamic but their GFP-Pcm1 exchanges with the cytosol are slow. Such results are consistent with the viscous liquid-like property of the condensates as revealed by their slow fusion kinetics (Fig. 7b; Supplementary Fig. 7a). Due to the limitation of our microscope system, we are unable to photobleach a part of the condensates to estimate the internal fluidity. Despite this, the slow fusion and limited turnover of the entire condensates suggest similar slow material exchanges within the condensates.

We also performed FRAP assays on the liquid droplets of Pcm1M and observed their highly dynamic protein exchanges with the milieu (Fig. 7i,j), consistent with their rapid fusion kinetics (Fig. 7h). The distinct phase separation abilities of different Pcm1 regions into highly dynamic liquid droplets (Pcm1M) or relatively static hydrogels (Pcm1N and Pcm1C) are consistent with the physical properties of the FGM condensates.

Minor comments:

1. Figure 1e looks as though a black smear has been placed over the top left corner to add in an XY axis, this should be removed and the axis placed accordingly on image so image is not altered in such a way

Response:

We are sorry for the confusion. During FIB-SEM, a protection layer of platinum is initially deposited on the surface of the sample block. The black smear was actually the leftover of the platinum layer. To avoid the confusion and allow clear examination of the FIB-SEM results, we have provided the 3D reconstructed image as a movie in the revised manuscript (Supplementary Movie 1).

2. Figure 5 is loaded with images, the small insets on sides of main images are in gray scale and with no labelling of which protein each gray scale image is for, suggestion for colors to be kept similar between main images and small cropped images or a clarification in legend or in figure of what each gray scale image is showing.

Response:

We appreciate the suggestion because we always find it challenging to clearly present images of multiciliated cells. In the initial manuscript, we used the colored arrowheads to indicate the original RGB channels of the grayscale insets. In the revised manuscript, we have used colored lines to indicate the channels in Fig. 5, 6 and Supplementary Fig. 4-6.

Reviewer #2:

Review of Zhao et al., 2020

The authors provide a detailed description of the role of fibrogranular material in centriole assembly and motile cilia function in multiciliated cells. Fibrogranular material has been observed in multiciliated cells for many years by electron microscopy, but its function has not yet been explored. This manuscript provides strong evidence that Pcm1 is an essential component of fibrogranular material, and that the absence of fibrogranular material results in deuterosomes that are more numerous, but smaller in size.

Additionally, motile cilia assembled in cells without fibrogranular material have structural and functional defects. Overall, I felt this manuscript provided an excellent characterization of fibrogranular material and introduces novel components of this structure that will be of significant interest to the centriole and motile cilia fields. The manuscript is clearly presented, and the quality of the data is generally very high. Below, I make a number of recommendations that should help improve this already excellent work. Many of these changes can be made by modifying the text or with additional analysis of the available data.

Response:

We thank our reviewer for appreciating our efforts. We have included the requested quantification results in the revised manuscript and addressed other concerns. We hope our

reviewer find that the revised manuscript is significantly improved over the previous one. We would also like to thank our reviewer for helping us to improve the manuscript.

Major points:

1. On lines 104-105, the authors write, “Interestingly, deuterosomes still formed massively and produced procentrioles in the Pcm1-depleted cells (Fig. 2c)”. However, there was no quantification of centriole or cilia number in the Pcm1-depleted cells. This reviewer believes it is essential to know if Pcm1-depleted cells amplify similar numbers of centrioles to control cells.

Response:

Following the requests, we quantified the basal body numbers. As mEPCs fixed at day 3 were still undergoing the centriole amplification, we used mEPCs at day 10, at which time they have fully matured into multiciliated cells (Fig. 3), to assess the eventual basal body numbers. As it is not practical to clearly count the number of multicilia in images of light microscopes, we chose to only count the number of basal bodies per cell and observed that the depletion of Pcm1 did not markedly affect the basal body production in mEPCs (Fig. 3d, revised manuscript). Similar results were obtained in mTECs when we depleted Pcm1 through an adenovirus-mediated RNAi (Supplementary Fig. 3d and 3e), consistent with the previous report (Vladar et al., 2007).

2. Related to point 1, the authors’ claim that deuterosomes still form following Pcm1 depletion is based on Cep152 immunofluorescence staining. However, it would be very helpful to stain deuterosomes directly using an antibody against Deup1. This will allow direct characterization of deuterosomes in control and Pcm1-depleted cells.

Response:

We thank our reviewer for the comment. We commonly use Cep152 as deuterosome marker in the manuscript because its antibody (chicken) is compatible with others in multiplex immunostaining. Our Deup1 antibody was produced in rabbit. Following the request, we co-stained Pcm1 with Deup1 in control and Pcm1-depleted mEPCs and have provided representative 3D-SIM images and quantification results on deuterosome number and size in Supplementary Fig. 2.

3. In figure 2g-h, the authors quantified deuterosome number and size and found that in Pcm1-depleted cells, deuterosome number increased while size decreased. It would be useful to also quantify if the number of procentrioles associated with each deuterosome decreases in Pcm1-depleted cells.

Response:

Following the request, we quantified the number of procentrioles per deuterosome and have presented the results in Fig. 2f as dot plots in the revised manuscript. Indeed the procentriole number per deuterosome decreased in the Pcm1-depleted cells. This is consistent with our

previous observations that smaller deuterosomes usually contain fewer procentrioles (Zhao et al., 2013; Zhao et al., 2019).

We have also changed the box plots in Fig. 2g and 2h of the initial manuscript into the more straightforward dot plots (Fig. 2d and 2e, revised manuscript). We found that some n values were incorrectly numbered in the two panels and have corrected them. We have also corrected the position of the MW marker for Odf1 in the immunoblot and accordingly cropped out the excessive blank region (Fig. 2b).

4. The labels in figure 2 are confusing. The quantifications in Figures 2g and 2h correspond to the siRNA experiments in figure 2d. Therefore, they should be relabeled as 2e and 2f, respectively. The rescue experiments shown in figure 2e and 2f should come last in the figure.

Response:

We are sorry for the confusion. The results in Fig. 2g and 2h of the initial manuscript actually correspond to Fig. 2c. The original arrangement was intended to fit the flow of the text. To avoid confusion, we have moved these two panels to Fig. 2d and 2e in the revised manuscript and modified the text accordingly.

5. The rescue experiment in figure 2e,f would benefit from more characterization. I appreciate that TEM analysis on the Pcm1-rescue cells would be challenging due to the low transduction efficiency of the adenovirus expressing the rescue construct. However, the authors could test if the rescue construct is able to rescue deuterosome size and number. Centriole number in MCCs with rescued Pcm1 should also be reported.

Response:

We thank our reviewer for the comments. Following the requests, we quantified deuterosome numbers and sizes in Pcm1-treated mEPCs expressing GFP-Pcm1 or Centrin1-GFP (negative control) and have presented the results in Fig. 2j and 2k, respectively, in the revised manuscript. The expression of GFP-Pcm1 indeed significantly restored the number and size of deuterosomes as compared to Centrin1-GFP.

We chose not to quantify the centriole numbers in the rescue experiments but instead measured ciliary beat frequencies in day-10 mEPCs (Fig. 3j). As the depletion of Pcm1 does not affect the basal body production in mEPCs and mTECs (Fig. 3b, 3d and Supplementary Fig. 3d and 3e) (Vladar et al., 2007), it does not appear to be logical to quantify the centriole numbers in the rescue experiments. Furthermore, as explained in our response to the major point #1, mEPCs fixed at day 3 were still undergoing the centriole amplification and therefore inappropriate for the purpose to assess the eventual basal body numbers. The use of Centrin1-GFP as negative control in the rescue experiments further complicated the situation due to its potential influences on both production and immunostaining of procentrioles/basal bodies. By contrast, the quantification results on beat frequencies (Fig. 3j) serve as better evidence for the rescue effect of GFP-Pcm1 than the centriole numbers. We hope that our reviewer would agree with us on this.

6. It is important to provide more information on motile cilia function in Pcm1-depleted cells. In figure 3d, the authors quantify planar versus abnormal cilia beating patterns. In the materials and methods, the authors should clearly explain what qualifies as an “abnormal” beating pattern. If just one cilium moves in a circular motion, does that qualify the cell as “abnormal,” or must all cilia be beating in a circular motion? In addition, cilia beat frequency should be measured in the control and Pcm1-depleted cells as an additional determination of cilia function.

Response:

We are sorry for missing the information. As it is impractical to trace the movement of every cilium, we scored multiciliated cells with their cilia beating predominantly in a back-and-forth pattern as “planar” and those predominantly containing immotile and/or irregularly-beating cilia as “abnormal”. We have included this information in the figure legend (Fig. 3f) and the Method section (Quantification and statistical analysis). We also quantified ciliary beat frequencies in both RNAi and rescue experiments and have presented the results in Fig. 3g and 3j, respectively. The beat frequencies were generally reduced and uneven in the Pcm1-depleted mEPCs as compared to the control cells (Fig. 3g). Such defects were largely rescued by GFP-Pcm1 but not Centrin1-GFP (Fig. 3j).

7. In Figure 7, the authors describe the condensate properties of fibrogranular material. However, in 7b, the GFP-Pcm1 droplets look completely different from the Pcm1 structures shown throughout the rest of the manuscript (for example, in figure 1, 2c, and 2f). This reviewer believes that the droplet-like localization of GFP-Pcm1 shown in figure 7b could be an artifact of overexpression: something that is common with overexpressed centrosome/centriole proteins. In the absence of monitoring Pcm1 behavior with an endogenous tag, I would strongly recommend removing figure 7b from the manuscript as this data could be very misleading. Instead, the authors should focus on the in vitro phase separation data supported by the disordered regions of the protein.

Response:

We are sorry for having not clearly explained the results. The images in Fig. 7b (and Supplementary Fig. 7) were taken from living cells with a spinning disk microscope, whose optical resolution is not comparable to 3D-SIM. They cannot display as many details as the 3D-SIM images of the fixed cells. The FGM condensates marked by GFP-Pcm1 thus appear as bright speckles in the live cell imaging. In addition, the pre-extraction procedure that we used for optimal immunostaining of fixed cells may also contribute to the difference by removing soluble fractions. When mEPCs expressing GFP-Pcm1 were imaged with 3D-SIM in the rescue experiments, GFP-Pcm1 displayed similar FGM localizations (Figure 2i, revised manuscript) as the endogenous Pcm1 (Supplementary Fig. 1a).

To further clarify this, we fixed mEPCs co-expressing GFP-Pcm1 and SNAP-Deup1 as in Fig. 7a at day 3 and imaged them with 3D-SIM. The subcellular localizations of GFP-Pcm1 (**Fig.**

2 for reviewers) are very similar to those of GFP-Pcm1 in the Pcm1-depleted mEPCs (Fig. 2i) and of endogenous Pcm1 (Supplementary Fig. 1a). The FGM networks marked by GFP-Pcm1 also enriched endogenous Cep131 (**Fig. 2 for reviewers**). Furthermore, deuterosomes marked by SNAP-Deup1 in these cells also generated procentrioles (**Fig. 2 for reviewers**). Therefore, we have indicated that we used a spinning disk microscope for the live cell imaging in the main text of the revised manuscript to avoid misunderstanding.

Fig. 2 for reviewers

Furthermore, we performed FRAP assays on the GFP-Pcm1 condensates and the liquid droplets of His-GFP-Pcm1M and have presented the results in Fig. 7c-7d and 7i-7j of the revised manuscript. We observed that the FGM condensates displayed slow but substantial exchanges of GFP-Pcm1 with the cytosol (Fig. 7c-7d), further indicating that the GFP-Pcm1 condensates are not aggregates formed by misfolded protein as suspected by our reviewer. In comparison, the liquid droplets of Pcm1M displayed highly dynamic protein exchanges with the milieu (Fig. 7i,j), consistent with their rapid fusion kinetics (Fig. 7h). The distinct phase separation abilities of different Pcm1 regions into highly dynamic liquid droplets (Pcm1M) or relatively static hydrogels (Pcm1N and Pcm1C) are consistent with the physical properties of the FGM condensates.

Minor points

1. Detailed methods for ependymal cell cultures and mTEC cultures should be described.

Response:

We have included the detailed methods in the Materials and Methods section of the revised manuscript.

2. In figure 1b-c, the authors show that throughout stages of differentiation, the number of Pcm1 condensates decreases, but their size increases. If possible, this could benefit from quantification.

Response:

We have included the quantification results in Fig. 1d and 1e.

3. The representative TEM images in figure 2d for control and Pcm1-depleted cells look very different. The deuterosomes are much darker in the Pcm-i1 and the procentrioles are not easy to distinguish. Can the authors comment on this apparent difference? Also, a representative 1 μ m image should be shown for the control sample to match the Pcm-i1 sample.

Response:

The difference might be due to variations in regional diffusion of the staining reagents, such as OsO₄ and uranyl acetate, during the EM sample preparations. To avoid confusion, we have replaced the control TEM image with one that has similar contrast as the Pcm1-i1 sample and presented it following our reviewer's request (Fig. 2g).

4. Relating to figure 3, in lines 148-150 in the text the authors state "In addition, 53% of ciliary transition zones (n = 109) and 52% of the basal bodies (n = 68) in the Pcm-i1-treated cells also displayed defects in microtubule organization, comparing to the 100% normal ultrastructure (n \geq 26) in the control cells (Fig. 3k,l)." The authors should show the quantifications for the transition zone and basal body sections of the cilia similar to how they showed these quantifications for the axoneme regions in figure 3j.

Response:

We have presented the quantification results in Fig. 3p in the revised manuscript as requested. We found that a basal body in the Pcm-i1 sample was accidentally counted twice so we have corrected the n number to 67.

Reviewer #3:

This study by Zhao and colleagues investigates the role of electron dense fibrogranular material (FGM) in multiciliated cells. They demonstrate that the centriolar satellite protein PCM1 is a core component of FGMs, and helps to form these structures via phase separation. The authors show that FGMs are distinct from deuterosomes and centrioles, but can localize and concentrate a number of centriole and cilia-associated proteins. Proteomic analysis of FGM proteins identified a number of centriolar and ciliary proteins, in addition to other cellular components. Disruption of FGMs via depletion of PCM1 in multiciliated cells caused abnormal centriolar targeting of FGM client proteins, which then resulted in altered deuterosome size, number, and distribution. Even though cilia formation was not impacted, the cilia showed structural defects that disrupted their motility. In sum, the authors conclude that FGMs in multiciliated cells help to concentrate deuterosome and centriole-related proteins, necessary for proper assembly of basal bodies and motile ciliary axonemes.

Overall, the story is very intriguing and somewhat consistent with prior data. However, I believe the manuscript suffers from a number of experimental flaws that makes it difficult for this reviewer to accept the interpretations of the results presented in this study. As such, I do not believe it is ready for publication in its current form.

Response:

We thank our reviewer for finding our work interesting. We have clarified the concerns in our responses and modified the manuscript accordingly during the revision. We hope our reviewer find that the revised manuscript is significantly improved over the previous one. We would also like to thank our reviewer for helping us to improve the manuscript.

Major concerns:

- One of the main issues is that the authors switch between airway multiciliated cells and ependymal multiciliated cells throughout the paper, performing different assays in each (detailed more below). Although this is presented as a strength, it is in fact a weakness because the biology of these two cell types is not the same, even though both are multiciliated cells. In recent years it is becoming increasingly evident that the kinetics of centriole amplification, centriole and cilia number, and organization of these structures is different between these two cell types. This is also evident when considering human disease phenotypes: there are many mutations that impact ciliary motility where patients present with respiratory defects (e.g. primary ciliary dyskinesia) due to defects in airway multiciliated cells, but without any ependymal cell dysfunction. Similarly, there are mutations that impact ependymal ciliary motility (causing hydrocephalus) that do not cause respiratory defects, and those patients have normal motile cilia function in airway multiciliated cells. Thus, at the cellular and organ level there are subtle differences in the biology of these cells, therefore correlating findings from one cell type to the other is not accurate in many cases.

Response:

We appreciate the comments and agree with our reviewer that ependymal and airway multiciliated cells are not identical despite of many shared properties. To clarify whether the major phenotypes of *Pcm1* depletion are common in both cell types, we depleted *Pcm1* in mTECs and have presented the results in the revised manuscript. We observed similar influences on deuterosomes and ciliary motility in mTECs (Supplementary Fig. 3a-3c, 3f, and 3g) as in mEPCs. Similar to the *Pcm1* depletion in mEPCs, the *Pcm1* depletion in mTECs also did not markedly affect the basal body numbers (Supplementary Fig. 3d and 3e). The depletion of *Pcm1* also caused premature centriolar targeting of Cep135 and elongated Cep131 streak in mTECs (Supplementary Fig. 6).

As our reviewer noticed, the distribution patterns of FGMs in the two cell types are not identical because we have confirmed that FGMs indeed lack the nuclear association in mTECs

(Supplementary Fig. 1b) in contrast to mEPCs (Supplementary Fig. 1a). The difference implies that the FGM has cell-specific functions. We speculate that an mEPC-specific protein or complex may tether the FGMs to the nuclear surface. In addition to recruiting some deuterosomes to the nuclear surface (Al Jord et al., 2017), the nucleus-tethered FGMs might contribute to the translational polarization of mEPC multicilia, a feature nonexistent for mTEC multicilia (Ohata & Alvarez-Buylla, 2016; Brooks and Wallingford, 2014). In the revised manuscript, we have similarly discussed these issues in the Discussion section.

- This issue is highlighted in Fig 1 and Supp Fig 1: in Fig 1 the authors do a convincing job of demonstrating that PCM1-containing FGMs are DISTINCT from deuterosomes and centrioles in airway multiciliated cells - using a combination of SIM and electron microscopy, they show that these FGMs form circular-looking structures by SIM that are adjacent to, but not localized to, either deuterosomes and centrioles (this point is emphasized throughout the paper). This is also confirmed by immune-gold labelling and TEM analysis of FGMs. However, when the authors try to confirm this finding in ependymal MCC, it is very obvious that PCM1-containing structures are localized differently in these cells. At almost every stage of centriole amplification and differentiation, the distribution of PCM1 and its relative position to deuterosomes and centrioles is different in the two cell types. There is a lot of PCM1 associated with both deuterosomes and procentrioles, and they are distributed in more puncta than the strictly circular looking FGMs in airway MCC. The authors use arrows to show a few examples of PCM1 not localizing to deuterosomes or procentrioles, yet the majority of PCM1 is in fact associated with those structures – I can clearly see many PCM1 positive deuterosomes and procentrioles, as well as the nuclear envelope enrichment (again not seen in airway MCC). Hence, the conclusion that PCM1 is in FGMs and generally is distinct from deuterosomes and newly forming centrioles is inaccurate. This is all important because the functional studies using depletion of PCM1 are all subsequently performed in ependymal MCC, where this is clearly not true.

Response:

We are afraid that our reviewer misunderstood some of our points. We actually have similar observations as our reviewer that deuterosomes are associated with FGMs. Our reviewer might have missed the related information in the manuscript, such as the title of Figure 7 (FGMs form through Pcm1 phase separation and tightly associate with deuterosomes) and the statement in the abstract (FGMs also tightly contacted with or even enwrapped deuterosome-procentriole complexes).

In our manuscript, we only consider a protein whose localization resembles those of deuterosomal Cep152/Deup1 in 3D-SIM as a deuterosome-localized protein. Taken the images of Supplementary Figure 1a (revised manuscript), in which our reviewer “clearly see many PCM1 positive deuterosomes”, as examples, Pcm1 is actually not seen to clearly localize on the deuterosomes (please refer to **Fig. 3a for reviewers**, arrows). Even if some Pcm1 puncta may

look partly overlapped with a portion of the deuterosomes, we do not consider this as an evidence of deuterosome localization. By contrast, taken the two control mEPCs in stages II and III in Supplementary Figure 5c (revised manuscript) as examples, Cep215 displayed clear deuterosome localization (please refer to **Fig. 3b for reviewers**, arrows). We therefore suspect that Cep215 and Pcnt, which displayed mild deuterosome localization (Supplementary Fig. 5d), might mediate the FGM-deuterosome interactions.

Fig. 3 for reviewers

The FGM appears to be complicated dynamic networks composed of sporadic fibrous granules (FGs), loose FG arrays, and tight FG condensates. FGMs in both mTECs and mEPCs contained these forms and enriched similar components. FGMs in mEPCs also formed large condensates (Fig. 4a-4b) that are comparable in size with those in mTECs (Fig. 4d and note that the scale bar length is longer than that in Fig. 4a-4b). Possibly their nuclear surface-associations limited the formation of as many discrete condensates as in mTECs.

As to the Pcm1-centriole relationship, previous studies in cycling cells (e.g., Dammermann & Medes, 2002; Tollenaere et al., 2015) indicate that Pcm1 is enriched in centriolar satellites but is not an integral component of the centriole. Similarly, Pcm1 puncta can be found in the vicinity of procentrioles or basal bodies in multiciliated cells but centriolar Pcm1 localization is not observed (e.g., Fig. 1 and Supplementary Fig. 1) (Kubo et al., 1999; Vladoar & Stearns, 2007), in sharp contrast to the typical components of procentrioles or basal bodies (Fig. 5 and Supplementary Fig. 4).

- Next, the authors deplete PCM1 to study FGM function, but they do this only in ependymal MCC using RNAi. They note a change in deuterosome number and size, but not centriole number or ciliogenesis (as was shown previously). Importantly, they note a

defect in basal body morphology (which is not quantified) and ciliary ultrastructure. It makes sense to me that loss of PCM1 might cause centriolar defects in ependymal MCC, since the protein is associated with the newly forming basal bodies. But what about in airway MCC where it is distinct from these structures, a major point of Fig1? For this reviewer to be convinced regarding the observed defects upon PCM1 depletion, the authors will need to inhibit the formation of FGMs in airway MCC and show the same phenomenon (basal body and cilia ultrastructural defects, abnormal motility) occurs there. Otherwise, the interpretation of this (and subsequent) experiment is inaccurate.

Response:

Our reviewer might have missed the descriptions on the quantification results for basal body morphology in the main text. To avoid confusion, we have presented the quantification results on basal body and transition zone in the revised manuscript (Fig. 3p).

To clarify the situations in mTECs, we depleted Pcm1 in mTECs with an adenovirus-expressed shRNA whose targeting sequence is identical to that of Pcm-i1 and observed similar influences on deuterosomes and ciliary motility (Supplementary Fig. 3a-3c, 3f, and 3g) as in mEPCs. Similar to the Pcm1 depletion in mEPCs, the Pcm1 depletion in mTECs also did not markedly affect the basal body numbers (Supplementary Fig. 3d and 3e). Furthermore, the depletion of Pcm1 also caused premature centriolar targeting of Cep135 and elongated Cep131 streak (Supplementary Fig. 6). Therefore, the FGM has similar functions in these aspects in both mEPCs and mTECs.

The limited viral infection efficiency unfortunately precluded the requested EM analysis in mTECs. Although we identified the adenovirus-infected mTECs through the infection marker GFP-Centrin1 in light microscopy (Supplementary Figs 3, 6), we were unable to distinguish them from the uninfected cells in EM samples for convincing EM analyses. As we already performed detailed EM analyses in mEPCs (Fig. 3k-3p), we hope that our reviewer would agree that this piece of data is not essential to the manuscript.

- The authors then perform proximity labeling proteomics using a PCM1 interactor, Cep131, and do this in ependymal MCC. They identify a host of proteins, some of which are associated with centrioles and cilia. To make the case that these centriolar proteins are associated with FGMs, they go back to airway MCC and do localization studies. Here again the same issue presents itself: for example, they suggest that Cep135 (a core cartwheel associated protein essential for procentriole nucleation) and Cep120 (a core centriolar protein necessary for their growth and elongation) are both found enriched in FGMs and are distinct from deuterosomes and procentrioles. Yet this is contradictory to the many published papers (including previously from this lab) showing these proteins are in fact associated with procentrioles during centriole amplification in airway MCC (also noted in Fig. 5). How are proteins, such as Cep135 and Cep120, that are normally enriched on procentrioles during their formation, and which has been shown by multiple groups, now be excluded from procentrioles during centriole amplification?

Response:

Our reviewer apparently misunderstood the results and points. We never intend to say that these proteins no longer localize to centrioles. Our reviewer might have missed the diagrams in Fig. 5, which illustrate when and where these FGM client proteins localize to centrioles.

- Fig 5 – since the identification of these centriolar “client” proteins was done in ependymal MCC using proximity labeling with Cep131 as bait, why did the authors move to airway MCC to validate their localization to FGMs?! I suspect it’s because of the same issues described above.

Response:

We actually validated FGM localizations of the client proteins in both mTECs and mEPCs. We present the mTEC results in detail because the numerous discrete condensates in mTECs allow clear identification of FGM components (Fig. 4d and Supplementary Fig. 4). To avoid redundancy, in mEPCs we only present detailed FGM localizations of Cep131 (Fig. 4a) and the APEX2-Cep131 fusion protein-labeled substrates (Fig. 4b), the majority of them are Cep131 (Fig. 4c). The tight correlations of the other client proteins with FGMs marked by Pcm1 can be clearly visualized in the control mEPCs presented in Fig. 6a-6b and Supplementary Fig. 5a-5d. Furthermore, their FGM localizations disappeared in the Pcm1-depleted cells (Fig. 6a-6b and Supplementary Fig. 5a-5d). Only Rootletin tended to form numerous fibrous structures in mEPCs (Supplementary Fig. 5e) but we sometimes observed its enrichment in FGM condensates as well.

As we already present the FGM localizations of Cep131 in mEPCs (Fig. 4a-4b), we chose not to present its mTEC localizations in Fig 4d to avoid redundancy. In the revised manuscript, such information can be visualized from control mTECs in Supplementary Fig. 6b.

- Although I am not as familiar with phase separation dynamics, most of the studies I have seen show much higher kinetics in rates of fusion/fission events. The live-cell imaging studies (Fig 7b) show these events happening on the scale of 20-25 minutes. Is this normal for phase separated structures?

Response:

We appreciate the question of our reviewer. Numerous publications indicate that phase separated condensates do have different physical properties (e.g., reviews by Banani et al., 2017 and Shin & Brangwynne, 2017). Gel-like structures, for instance, are relatively static, whereas liquid droplets formed by different protein(s) can exhibit different extent of dynamics. Sometimes liquid droplets can also gradually age into gel-like condensates. The slow fusion kinetics suggests that the condensates are viscous, which is further confirmed by their slow exchange of GFP-Pcm1 with the cytosol in the newly provided FRAP results (Fig. 7c and 7d). By contrast, the rapid fusion kinetics of the liquid droplets of GFP-Pcm1M (Fig. 7h) suggests that they are of high fluidity. Consistently, the molecule exchanges are quite frequent as demonstrated in the newly provided FRAP results (Fig. 7i and 7j). Therefore, the viscous property (Fig. 7b-7d) and the porous morphology (please refer to Fig. 1) of the FGM condensates are consistent with the

distinct phase separation abilities of different Pcm1 fragments into either liquid droplets (Pcm1M) or hydrogels (Pcm1N and Pcm1C) *in vitro* (Fig. 7e-7j).

REVIEWERS' COMMENTS

Reviewer #1 (Remarks to the Author):

My concerns have been addressed and i now support publication of this paper.

Reviewer #2 (Remarks to the Author):

The review experiments were extensive and satisfied my concerns, therefore I recommend this manuscript for publication.

Minor comment

- The authors state that depletion of Pcm1 in mTECs results in phenotypes similar to those observed when Pcm1 is depleted in mEPCs. However, the authors saw ~50% increase in deuterosome number and ~25% decrease in deuterosome size in mEPCs, while in mTECs these differences were far less pronounced. The claim that Pcm1 functions are conserved in multiciliated cells is therefore potentially an overstatement of the data. I recommend revising the text to state that depletion of Pcm1 in mTECs had a more mild effects than in mEPCs.

Reviewer #3 (Remarks to the Author):

This is a revised manuscript by Zhao and colleagues, which investigates the role of PCM1-containing electron dense fibrogranular material (FGM) in multiciliated cells. The revised manuscript addresses a number of weaknesses identified in the original submission, including the inclusion of additional quantifications from various experiments, as well as clarifications in the text. As such, I believe it is suitable for publication with a few minor edits. With regards to concerns that I had raised:

- My main concern was about all of the depletion experiments being performed in only one cell type (mEPC). The authors have now performed depletion of PCM1 in mTECs and confirmed the observations with regards to deuterosome number and distribution, centriole amplification, centriole ultrastructure, and ciliary motility defects. These results look convincing, confirm the observations made in mEPCs, and solidify the conclusions with regards to roles of FGMs in MCC.

- The authors have made an effort to distinguish between the localization and distribution of PCM1-containing FGMs in mTECs and mEPCs, by expanding on these in the text and discussion. However, there is still some confusion throughout the story. For example, in the initial parts of the paper (e.g. Fig 1) they highlight the fact that although PCM1-containing FGMs are "mingling" with deuterosomes, they are in fact distinct structures that neighbor them. Then later in the story, it appears that deuterosomes may themselves be surrounded by FGMs, in liquid-like structures. To me, this seems counter intuitive. Maybe it is simply semantics, but I feel this point needs to be clarified further... are they distinct structures that only interact at certain stages of centriologenesis? They obviously interact functionally, since loss of PCM1 causes the dispersion of deuterosomes and their size/number.

- Depletion of PCM1 in mEPCs and mTECs resulted in dramatic alteration in deuterosome number, size and distribution. Yet, it is very interesting that basal body formation and number was ultimately unaffected. Did the authors look for any alterations in cell size? As noted on page 6 "The crop size that we used was usually sufficient to cover all the deuterosomes of a control mEPC but frequently unable to do so for Pcm1-depleted mEPCs". I wonder if there was a change in cell size/surface area in PCM1-depleted mEPC or mTEC? A recent study (Nanjundappa et al, eLife 2018) showed a relationship between cell size and basal body number, and that eliminating parental centrioles leads to

overproduction of deuterosomes (reminiscent of what appears here) which compensate for their loss, leading ultimately to formation of normal basal body number. Similarly, depletion of both parental centrioles and deuterosomes simultaneously causes normal centriole amplification and abundance (Mercey et al, Nature Cell Biology, 2019). These two studies suggest that a different cellular property (ie. cell size and/or surface area) may be more important than parental centrioles, deuterosomes, and now possibly FGMs in regulating centriole-cilia formation and number in MCC. Since loss of FGM integrity (upon depletion of PCM1) in this study shows similar findings, I believe it is important for the authors to consider changes in their cell (or surface area) size that may be compensating for loss of FGMs. At the very least, the authors should cite these studies and discuss them in this context.

Response to reviewers' comments

Reviewer #1:

My concerns have been addressed and i now support publication of this paper.

Response:

We thank our reviewer for appreciating our effects.

Reviewer #2:

The review experiments were extensive and satisfied my concerns, therefore I recommend this manuscript for publication.

Minor comment

- The authors state that depletion of Pcm1 in mTECs results in phenotypes similar to those observed when Pcm1 is depleted in mEPCs. However, the authors saw ~50% increase in deuterosome number and ~25% decrease in deuterosome size in mEPCs, while in mTECs these differences were far less pronounced. The claim that Pcm1 functions are conserved in multiciliated cells is therefore potentially an overstatement of the data. I recommend revising the text to state that depletion of Pcm1 in mTECs had a more mild effects than in mEPCs.

Response:

Following the request, we have modified the text as “we also observed increased deuterosome numbers and a mild decrease (4.4% on average) in deuterosome size in Pcm1-depleted cells as compared to mTECs expressing a control shRNA (shCtrl-i) (Supplementary Fig. 3a-c)” (page 7, 2nd paragraph, revised manuscript).

Reviewer #3:

This is a revised manuscript by Zhao and colleagues, which investigates the role of PCM1-containing electron dense fibrogranular material (FGM) in multiciliated cells. The revised manuscript addresses a number of weaknesses identified in the original submission, including the inclusion of additional quantifications from various experiments, as well as clarifications in the text. As such, I believe it is suitable for publication with a few minor edits. With regards to concerns that I had raised:

- My main concern was about all of the depletion experiments being performed in only one cell type (mEPC). The authors have now performed depletion of PCM1 in mTECs and confirmed the observations with regards to deuterosome number and distribution, centriole amplification, centriole ultrastructure, and ciliary motility defects. These results look convincing, confirm the observations made in mEPCs, and solidify the conclusions

with regards to roles of FGMs in MCC.

Response:

We thank our reviewer for appreciating our effects.

- The authors have made an effort to distinguish between the localization and distribution of PCM1-containing FGMs in mTECs and mEPCs, by expanding on these in the text and discussion. However, there is still some confusion throughout the story. For example, in the initial parts of the paper (e.g. Fig 1) they highlight the fact that although PCM1-containing FGMs are “mingling” with deuterosomes, they are in fact distinct structures that neighbor them. Then later in the story, it appears that deuterosomes may themselves be surrounded by FGMs, in liquid-like structures. To me, this seems counter intuitive. Maybe it is simply semantics, but I feel this point needs to be clarified further... are they distinct structures that only interact at certain stages of centriologenesis? They obviously interact functionally, since loss of PCM1 causes the dispersion of deuterosomes and their size/number.

Response:

We appreciate the comments of our reviewer. As the resolution of spinning disk microscopy used for live imaging (Fig. 7b) did not allow us to determine whether the deuterosomes were anchored to the surface of the large foci or buried inside them, we have stated this in the revised manuscript (page 12, 1st paragraph) and stopped using the word “enwrap” in the revised manuscript. Instead, we have consistently used “associate”, “interact”, or their equivalents to objectively describe the relationship between deuterosomes and FGM condensates.

- Depletion of PCM1 in mEPCs and mTECs resulted in dramatic alteration in deuterosome number, size and distribution. Yet, it is very interesting that basal body formation and number was ultimately unaffected. Did the authors look for any alterations in cell size? As noted on page 6 “The crop size that we used was usually sufficient to cover all the deuterosomes of a control mEPC but frequently unable to do so for Pcm1-depleted mEPCs”. I wonder if there was a change in cell size/surface area in PCM1-depleted mEPC or mTEC? A recent study (Nanjundappa et al, eLife 2018) showed a relationship between cell size and basal body number, and that eliminating parental centrioles leads to overproduction of deuterosomes (reminiscent of what appears here) which compensate for their loss, leading ultimately to formation of normal basal body number. Similarly, depletion of both parental centrioles and deuterosomes simultaneously causes normal centriole amplification and abundance (Mercey et al, Nature Cell Biology, 2019). These two studies suggest that a different cellular property (ie. cell size and/or surface area) may be more important than parental centrioles, deuterosomes, and now possibly FGMs in regulating centriole-cilia formation and number in MCC. Since loss of FGM integrity (upon depletion of PCM1) in this study shows similar findings, I believe it is important for the authors to consider changes in their cell (or surface area) size that may be compensating for loss of FGMs. At the very least, the authors should cite these studies and discuss them in this context.

Response:

We thank our reviewer for the insightful comments. We agree that how MCCs manage to keep a relatively constant basal body number is an interesting question. As requested, we have discussed this issue and cited these publications in the Discussion section (last paragraph) of the revised manuscript.

As to the issue of cell size, the linear relationship between cell surface area and basal body number is reported for mTECs (Nanjundappa et al., 2019). Multicilia in mTECs are densely distributed on the entire apical surface. More cilia thus require a larger surface area to accommodate. By contrast, although mEPCs are usually much larger in size (surface area) than mTECs, they contain much less cilia than mTECs (please refer to Fig. 3b,d and Supplementary Fig. 3d,e for a comparison). Furthermore, multicilia in mEPCs are confined in a certain apical area (which is termed the transitional polarization) (Brooks and Wallingford, 2014; Ohata and Alvarez-Buylla, 2016). Therefore, whether the basal body number in mEPCs is also proportional to the surface area needs to be clarified in future studies. More importantly, as the authors have pointed out, even in mTECs, whether having a larger surface area results in the formation of more centrioles or a cell that forms a larger number of centrioles expand its surface area to accommodate them remains unclear (Nanjundappa et al., 2019). We thus feel that it is inappropriate to involve this issue in our study.

References:

- Brooks, E.R., and J.B. Wallingford. 2014. Multiciliated cells. *Curr Biol.* 24:R973-982.
- Nanjundappa, R., D. Kong, K. Shim, T. Stearns, S.L. Brody, J. Loncarek, and M.R. Mahjoub. 2019. Regulation of cilia abundance in multiciliated cells. *Elife.* 8.
- Ohata, S., and A. Alvarez-Buylla. 2016. Planar Organization of Multiciliated Ependymal (E1) Cells in the Brain Ventricular Epithelium. *Trends Neurosci.* 39:543-551.